# INTERPRETING THE INNER MECHANISMS OF LARGE LANGUAGE MODELS IN MATHEMATICAL ADDITION

## ABSTRACT

Large language models (LLMs) have achieved stunning performance on various language tasks, but remain as mysterious as a black box. Understanding the internal mechanisms of LLMs could contribute to the development of more transparent and interpretable LLMs. To this end, we take the first attempt to reveal a specific mechanism relating to how LLMs implement the reasoning task of a mathematical addition, *e.g.*, scenarios involving simple one-digit integer addition. Through comprehensive experiments, we find that LLMs frequently involve a small fraction of attention heads ($0.5\%$ of all heads) when implementing the addition task. Meanwhile, knocking out these frequently involved heads significantly degrades the LLMs' performance on the same task. Surprisingly, these key heads identified for a specific dataset exhibit outstanding generalizability across multiple datasets related to the mathematical addition task. Moreover, we find an intuitive phenomenon that knocking out these key heads could also affect the performance of LLMs on mathematical subtraction, which shares the same spirit with human behavior. Our work serves as a preliminary exploration into the mathematical prowess of LLMs, laying a solid foundation to reveal more intricate capabilities.

R$_{K8c8}$.Q1

## 1 INTRODUCTION

Large language models (LLMs) have experienced rapid advancements and shown impressive language understanding capabilities (Devlin et al., 2019; Brown et al., 2020; Chowdhery et al., 2022). Notably, LLMs exhibit emergent abilities (Wei et al., 2022b) that enable them to solve intricate reasoning tasks akin to humans, such as mathematical computations (Frieder et al., 2023b; Jie et al., 2022), chain-of-thought reasoning (Wei et al., 2022c; Kojima et al., 2022), few-shot prompting (Brown et al., 2020; Alayrac et al., 2022), etc. Despite these impressive characteristics, the underlying mechanisms behind LLMs yet remain as enigmatic as a black box. This is due to the complex and intricate non-linear interactions within densely-connected layers, which form the foundation of LLMs. Consequently, as the size of LLMs continues expanding, interpreting their behavioral patterns requires careful design, raising a compelling but challenging domain in the field of LLMs.

Recent research in mechanistic interpretability has been devoted to reverse engineering the computational processes of model weights, making them understandable to humans (Elhage et al., 2021; Geva et al., 2020; Olah, 2022; Meng et al., 2022). Comprehending these underlying mechanisms could contribute to predicting how the LLMs behave beyond their training data (Mu & Andreas, 2020). Additionally, understanding the internal mechanisms of LLMs can identify and rectify errors present in the model (Hernandez et al., 2021; Vig et al., 2020), as well as gain insights into the emergence of certain behaviors (Nanda & Lieberum, 2022; Barak et al., 2022; Wei et al., 2022a). The interpretability research aids in the advancement of more transparent and interpretable LLMs.

In this work, we take the first attempt to explore the reasoning capabilities of LLMs through the lens of mathematical addition problems, which is conducted on three publicly available LLMs: LLaMA2-7B (Touvron et al., 2023), Qwen-7B (Bai et al., 2023) and chatGLM2-6B (Du et al., 2022). Unlike typical language comprehension tasks, mathematical addition problems have straightforward text content and distinct correct answers, and they require reasoning and calculating rather than direct copying to derive the results. These characteristics enable us to gain insights into the models' mathematical inference abilities without interference from unrelated factors. To this end, we create datasets of various types of sentences that involve the addition logic. For example, "The war lasted

2 years from the year 1734 to the year 173_" implicitly includes the addition "2 + 1734 = 1736". These LLMs could provide answers with high confidence scores of over $80\%$.

To unveil how these models correctly complete the task (*e.g.*, "$2 + 1734 = 1736$" in the aforementioned case), we do a hard intervention (Pearl, 2009) on the transformer attention heads to validate their effects on the predicted logits. The findings show that only a small percentage (0.5%) of the attention heads significantly impact the model's performance. Namely, LLMs frequently involve these attention heads when completing the task. Subsequently, we knocked out these frequently-involved heads to validate the faithfulness of these heads. We found that the model performance decreases dramatically when the frequently-involved attention heads are knocked out, resulting in a decrease of $70\%$ in accuracy. In contrast, knocking out the remaining $99.5\%$ heads causes limited influence, *i.e.*, a decrease of $3\%$ in accuracy. All these observations hold for all three large language models.

Furthermore, apart from the data used to discover the important attention heads, our analyses demonstrate that these heads can be generalized to unexplored data formats of addition. This not only underscores the crucial role these discovered heads play in mathematical addition, but also emphasizes their remarkable ability to generalize. What's even more surprising is that this phenomenon extends beyond addition and also applies to subtraction. When these key heads are deactivated, the model's capability to perform subtraction is significantly diminished, dropping by a substantial $52\%$.

In summary, this work aims to delve into the inner mechanisms of LLMs and uncover the black-box through mathematical addition task. Our findings reveal a striking sparsity in the model's attention heads, with less than 0.5% exhibiting close correlations. Remarkably, the absence of these heads leads to a relatively notable decline in model performance, compared to the absence of randomly-selected heads. Moreover, these discovered heads demonstrate equal effectiveness in both new formats of addition and subtraction tasks. These intriguing phenomena are consistent across various large language models, establishing a robust basis for a comprehensive comprehension of LLMs.

## 2 BACKGROUND

**Interpretability for Mathematical Tasks.** Mathematical ability has long been a subject of interest in natural language processing (Kushman et al., 2014; Huang et al., 2016; Wang et al., 2017; Thawani et al., 2021). Some studies have investigated the mathematical abilities of LLMs (Frieder et al., 2023a; Saxton et al., 2019; Nogueira et al., 2021; Qian et al., 2023), but they mainly focus on explaining *what* these models can do rather than *how* they do it. The studies conducted by (Wu et al., 2023) scale the methods from causal abstraction to understand how Alpaca (7B) (Taori et al., 2023) follows the instruction in comparing two numbers. Hanna et al. (2023) provide a causal explanation about how GPT2-samll (0.1B) (Radford et al., 2019) implements the "greater-than" task, but only reveal simple phenomena limited by the small size of model and the lack of diversity in the dataset.

**Large Language Models (LLMs).** The LLMs utilized in this study comprise LLaMA2-7B (Touvron et al., 2023), Qwen-7B (Bai et al., 2023) and chatGLM2-6B (Du et al., 2022; Zeng et al., 2023). These models' weights are freely available and can be acquired from HuggingFace[1]. All of these models are decoder-only transformers equipped with multiple attention heads and a single MLP in each layer. LLaMA2-7B and Qwen-7B consist of 32 layers and 32 attention heads per attention layer, while chatGLM2-6B comprises 28 layers and 32 attention heads per attention layer. In this work, we concentrate on comprehending the behavior of attention heads, where we apply the notation of "$i.j$" to refer to the $j$-th head of the attention layer $i$.

**Transformer Architecture.** The input to the transformer is a combination of position and token embeddings in $\mathbb{R}^{N \times d}$, where $N$ is the number of tokens in the input and $d$ is the model dimension. Following the definitions in (Elhage et al., 2021), the input embedding serves as the initial value for the *residual stream*, which is read from and written to by all attention heads and MLPs. Focusing on individual heads, the $j$-th head in the $i$-th layer is denoted as $h_{i,j}$, and parametrized by four matrices: $W_Q^{i,j}, W_K^{i,j}, W_O^{i,j} \in \mathbb{R}^{d \times \frac{d}{H}}$, and $W_V^{i,j} \in \mathbb{R}^{\frac{d}{H} \times d}$. To simplify these parameters, we can express them as low-rank matrices in $\mathbb{R}^{d \times d}$: $W_{OV}^{i,j} = W_O^{i,j} W_V^{i,j}$ and $W_{QK}^{i,j} = W_Q^{i,j} (W_K^{i,j})^T$. The QK matrix is used to compute the attention pattern $A_{i,j} \in \mathbb{R}^{N \times N}$ for head $(i,j)$, while the OV matrix determines

---

[1]https://huggingface.co/

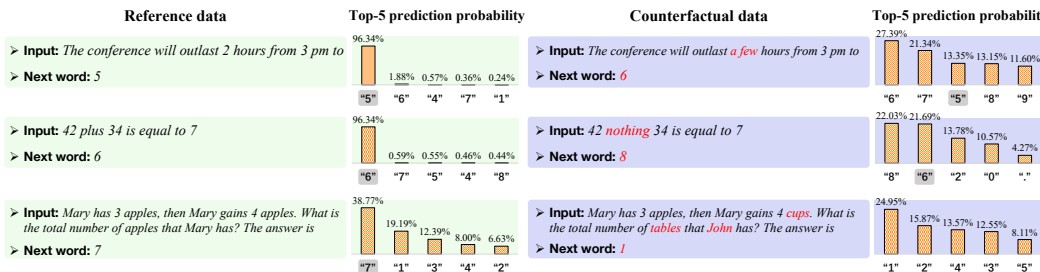

Figure 1: Examples of reference data (with addition logic) and counterfactual data (without addition logic). Given the input sentence, the results of next word prediction are provided by LLaMA2-7B.

the information written into the residual stream. At the end of the forward pass, a layer norm is applied before the unembed matrix $W_U$ projects the residual stream into logits.

**Task and Dataset.** We focus on a classic and widely encountered mathematical operation: addition. For example, the arithmetic logic of addition ($A + B = C$) might naturally appear in sentences like "The war lasted $\{A\}$ years from the year $173\{B\}$ to the year $173\{C\}$". The addition task is to predict the final token in the sentence to be the right answer $\{C\}$. Taking inspiration from the sentence styles and forms present in mathematical benchmarks of GSM8K (Cobbe et al., 2021) and MultiArith (Roy & Roth, 2015), we construct various types of templates as follows (see Appendix A for more templates):

1. "The `<event>` `<verb>` $\{A\}$ years from the year `<YYY>`$\{B\}$ to the year `<YYY>`$\{C\}$"

2. "$\{A\}$ plus $\{B\}$ is equal to $\{C\}$"

3. "`<name>` has $\{A\}$ `<object>`, then `<name>` gains $\{B\}$ `<object>`. What's the total number of `<object>` that `<name>` has? The answer is $\{C\}$"

We create a dataset for the addition task containing $100,000$ samples based on 20 templates with random single-token names, objects, events, and verbs. For example, we sample the `<YYY>` from $\{100, \cdots, 199\}$, $\{A\}$ and $\{B\}$ from $\{1, \cdots, 9\}^2$. To assess the performance of LLMs on the addition task, we measure the prediction probability of the $\{C\}$ token. For instance, considering the first sentence in Figure 1, the model provides a highly confident correct prediction with a probability of $96.34\%$. The average probability of correct predictions across the three models was $82\%$. In this study, we select the samples that the language models are able to predict correctly. We denote the sentences generated by this procedure as reference data using the notation of $X_r$.

Moreover, to meet the demand for perturbing component activation, we create another dataset comprising counterfactual sentences without the inclusion of logical addition, using the notation of $X_c$. As shown in Figure 1, the samples are generated following two core principles: (1) maintaining the grammatical structures derived from the $X_r$ templates; (2) substituting several crucial words responsible for the addition logic with irrelevant words. In this way, it allows for a direct reflection of the model's impact on the addition task, rather than being influenced by the sentence structure or syntax.

## 3 METHOD

Mechanistic interpretability aims to discover, understand, and validate the inner mechanisms of the model by reverse engineering model computation into human-understandable components. In this work, a directed acyclic graph (DAG) is reorganized to represent the computation of LLM concretely, as shown in Figure 2. In the graph, each node is a computation component, including attention heads, MLP layers, residual connections, and each edge represents the data flow that the output of the previous node will be transposed to the input of the later node. The input sentence is gradually processed node-by-node along with these edges to generate the next word prediction.

**Path Patching.** To discover the cause of the predicted answer, we employ the causal intervention technique known as *path patching* (Goldowsky-Dill et al., 2023; Wang et al., 2023). This approach

---

[2]For simplicity, the samples where $\{A\} + \{B\} \geq 10$ are not considered.

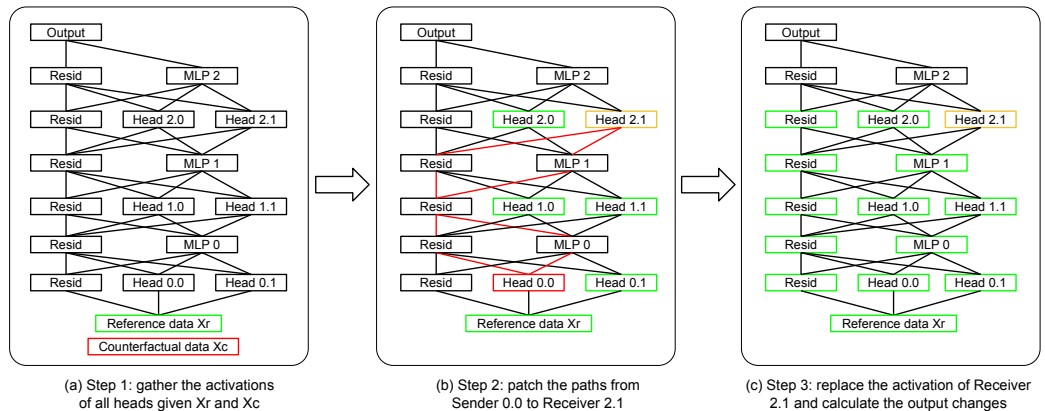

Figure 2: A case illustration of the method "path patching". It measures the importance of forward paths, that originate from Head 0.0 to Head 2.1, for the three-layer transformer in completing the task on reference data.

is highly effective in analyzing the causal relationship between two computation nodes (Sender → Receiver). This helps us determine whether Sender is the cause of Receiver, and the connections $\mathcal{P}$ between them are important for the model in implementing the task.

Specifically, the entire process of path patching is divided into three steps, as shown in Figure 2 where the node pair Sender → Receiver is set as Head 0.0 → Head 2.1. Firstly, given $X_r$ and $X_c$, the activations of all heads are gathered for preparation of the later perturbation. Then, the Sender is perturbated by the activation from $X_c$, which will be further propagated to Receiver along with $\mathcal{P}$. To ensure an independent observation of the impact from the Sender, $\mathcal{P}$ comprises the forward pathways through residual connections and MLPs except for other attention heads (*e.g.*, Head 0.1, 1.0, 1.1), represented by the red lines in Figure 2(b). Finally, to measure the impact of this perturbation, we output the final logits when the original activation of Receiver is replaced by the perturbated activation, as shown in Figure 2(c). Overall, the implementation of path patching can be summarized as:

1. Run forward pass to gather the activations of all heads given the reference data $X_r$ and counterfactual data $X_c$.

2. Run forward pass to record receiver head activation, while keeping all the heads frozen to their activations on $X_r$, except for sender head whose activation is set on $X_c$.

3. Run forward pass to measure the change of output logits, while replacing the activation of receiver head with the recorded value in step 2.

If there is a significant change in final logits, then the patched paths: Sender → Receiver are essential for the model in completing the task.

In this work, to identify the important heads contributing to the addition task, we scan through all heads as the Sender node denoted by $h$, and set the Receiver node as output $logits$, and measure the changes in the output logit of ground-truth token {C}. Pathways $h \to logits$ that are critical to the model's computation should induce a large drop in the logit of token {C} after patching. Notably, since the residual operations and MLPs compute each token separately (Elhage et al., 2021), patching the head output at the END position (*i.e.*, the position of the last token in the input sentence) is enough to measure the effects on the next token prediction.

**Knockout.** Explanations for model behavior can easily be misleading or non-rigorous (Bolukbasi et al., 2021; Wiegreffe & Pinter, 2019). To address this issue, we adopt a thorough approach to assess the importance of each node identified through path patching, while also confirming the insignificance of other nodes. For this purpose, we employ a knockout technique called *mean ablation* (Wang et al., 2023) to deactivate individual nodes and analyze their impact on model performance. Specifically, we replace the activation of nodes with their average activation across counterfactual data $X_c$ to remove the task-related information. By observing changes in model performance, we can verify the roles of these key heads.

## 4 EXPERIMENTS

To interpret the model computation into human-understandable components, we perform experiments as follows: (1) *discover* the important components in completing the addition task by path patching in Section 4.1; (2) *understand* the behavior of the newly identified components by examining their attention patterns in Section 4.2; (3) *validate* the faithfulness of the important components by knockout analysis in Section 4.3; (4) *transfer* to other related tasks to verify the working mechanism of discovered components to be generalizable in Section 4.4. Additionally, in Section 4.5, we take a further step to investigate whether there are any underlying factors in earlier layers that may affect model prediction through the identified components.

### 4.1 WHICH HEADS DIRECTLY AFFECT THE OUTPUT LOGITS?

To unveil how the model predicts the result token {C}, we search for attention heads that directly affect the model's output logits by patching the pathways: $h \to logits$.

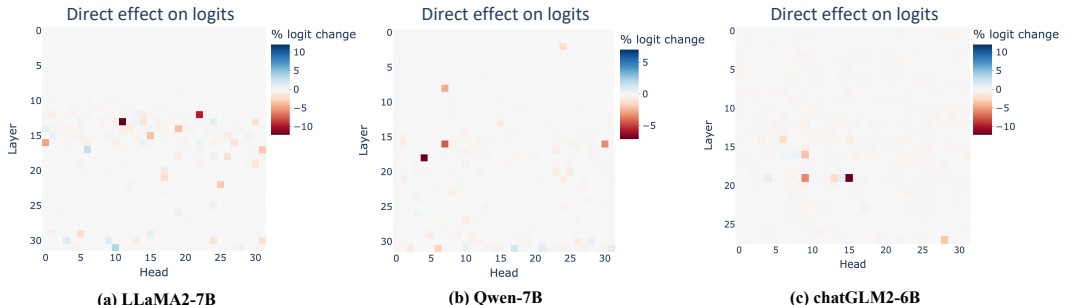

(a) LLaMA2-7B        (b) Qwen-7B        (c) chatGLM2-6B

Figure 3: We conduct path patching experiments on three different LLMs, by searching for each head $h$ directly affecting the logit of the right answer. For each head, a darker color indicates a larger logit difference from the original model before patching.

In Figure 3, we visualize the effect of each head according to the serial numbers of the heads and layers. This arrangement allows for a clear comparison of the causal impact of each head to the logit of ground-truth token {C}. The red squares indicate heads that have a significant positive impact on predicting the output token, while the blue squares represent heads that have a negative effect. From the overall visualization results, we observe that:

(i) Only a small number of heads have a noteworthy influence on the output. Specifically, when heads such as 13.11 or 12.22 in LLaMA2-7B is patched, there is a substantial decrease of 12.1% and 9.6% on the logit of token {C}, respectively, which highlights their positive contribution to the addition task. The same phenomenon can also be observed for the heads 18.4 and 16.7 in Qwen-7B, as well as the heads 19.15 and 19.9 in chatGLM2-6B. The sparse distribution of these key heads, which is consistent across different models, motivates us to explore their specific functionalities and characteristics. (More analysis of the key heads sparsity can be viewed in Appendix E.)

(ii) The discovered key heads are mainly located in the middle layers. In the case of head 13.11 in LLaMA2-7B, it exerts the most significant direct effect on the logit of token {C} through residual connections and MLPs, while the heads in subsequent layers (13-31) have minor direct effects. Furthermore, although the heads in earlier layers (0-12) may have lesser direct effects on the model output, they can still influence the model output indirectly through head 13.11. To provide more insights into this matter, we conduct additional investigations in Section 4.5.

For simplicity, in the following experiments, we primarily report the results of LLaMA2-7B, while the results of the other two models can be found in Appendix C and D.

### 4.2 WHAT IS THE BEHAVIOR OF KEY HEADS?

In order to better understand the "behavior" of the heads that have a significant impact on addition, we begin by analyzing their attention patterns. Our findings reveal that these heads exhibit a strong

focus on the numerical tokens {A} and {B} in the input sentences while being insensitive towards other tokens. On average, these heads assign an attention probability of 0.62 to numerical tokens across all samples of different template types. Drawing upon this observation, we propose two hypotheses regarding these heads: (i) they have specific attention towards number tokens, and (ii) this attention pattern remains consistent across diverse input contents.

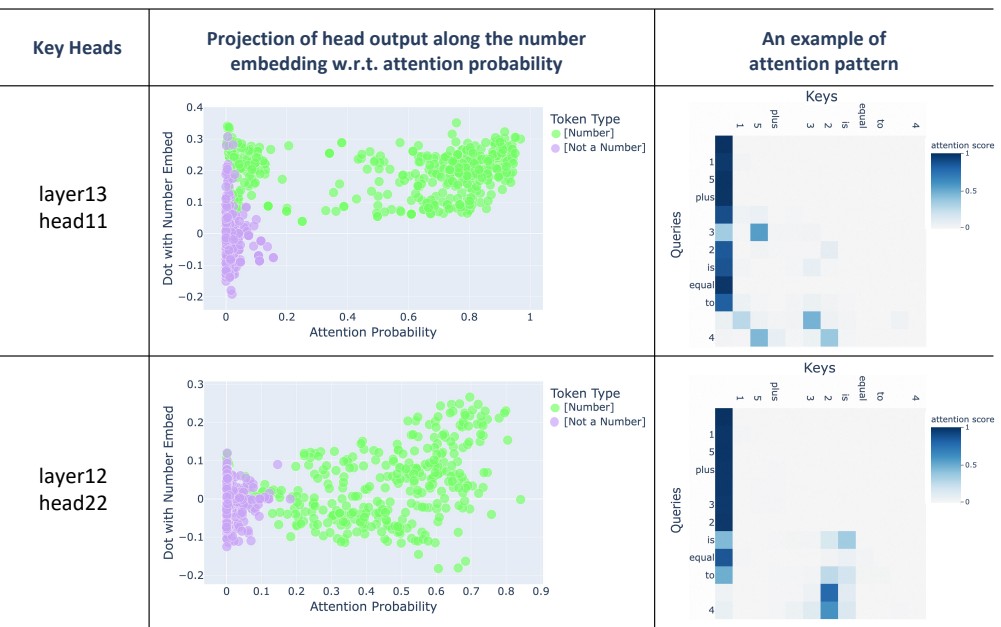

Figure 4: Left: two examples of key heads in LLaMA2-7B. Middle: we randomly sample 500 samples and show their attention probability and projection of the head output along number token or not, respectively. Right: we visualize the head's attention pattern to inspect the last row for the attention scores between Query token at the END position and each Key token.

To verify our hypotheses, we design experiments to test the heads' functionality. Let $W_U$ denote the unembedding matrix, $W_U[N]$ denote the corresponding unembedding vectors for the number tokens (*e.g.*, {A} and {B}), and $W_U[NaN]$ for tokens that are not a number (*e.g.*, "war", "apples", "Mary", etc.). We randomly sample 500 samples from reference data $X_r$ and scatter plot the attention probability against the inner product of vectors $\langle h_i(X), W_U[N] \rangle$ and $\langle h_i(X), W_U[NaN] \rangle$, measuring how much head $h_i$ on input $X$ is writing in the direction of the logit of the number tokens $N$ (A and B) and non-numeric tokens $NaN$, respectively. The results are shown in Figure 4: the heads (*e.g.*, 13.11, 12.22) present a much higher attention probability on number tokens than other tokens (0.73 vs. 0.08), with a much higher output in the direction of the number (0.21 vs. −0.05).

Furthermore, we visualize the attention patterns of these heads on different sentences. Since we target at exploring the next token prediction at the END position, we focus on the *last row* of each attention map to check the attention scores between Query END token and each Key token. In Figure 4, the two samples indicate that these heads specifically attend to the number tokens "5" and "2". Additional results in Figure 5 demonstrate that: (i) In different sentences involving addition 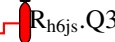 (samples 1-3), the heads primarily prioritize number tokens, regardless of their positions in the sentence. (ii) Even in sentences without any linguistic meaning (samples 4-6), the heads still prioritize attending to the number tokens.

The above results reveal a distinct working mechanism of the discovered key heads on number tokens, and verify the aforementioned hypotheses. More results from another two models in Appendix C further validate this claim.

### 4.3 VALIDATION OF KEY HEADS.

To fully validate the claimed functionality or effect of the discovered key heads, we perform additional checks by knocking out these heads. In Figure 6a, all heads are sorted in a certain order

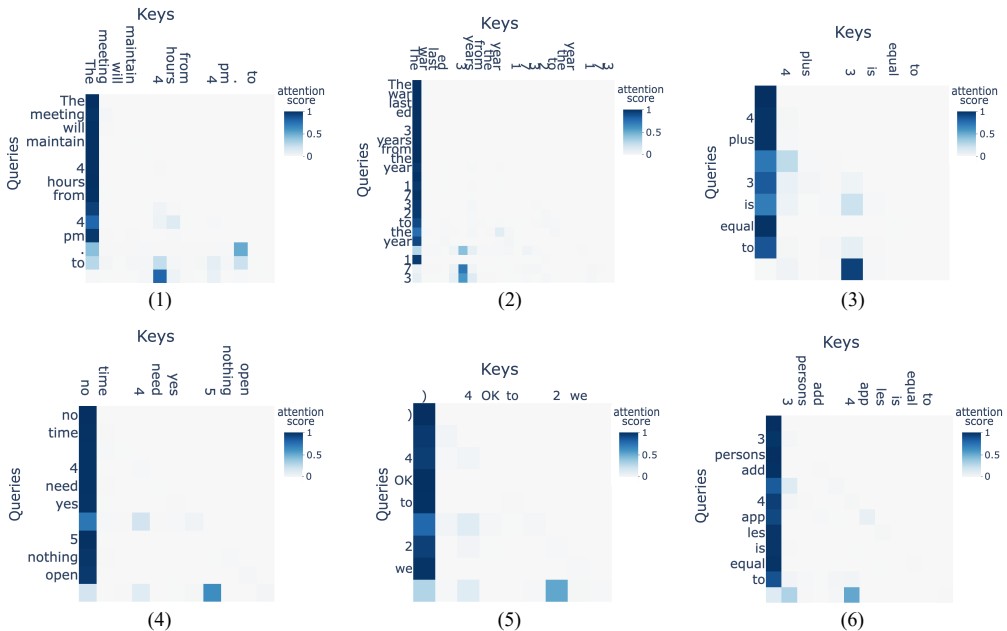

Figure 5: The attention patterns of key heads (*e.g.*, 13.11 in LLaMA2-7B) on different samples. The (1-3) samples are sentences with the addition logic in different templates, while the (4-6) samples are randomly constructed sentences with no linguistic meaning.

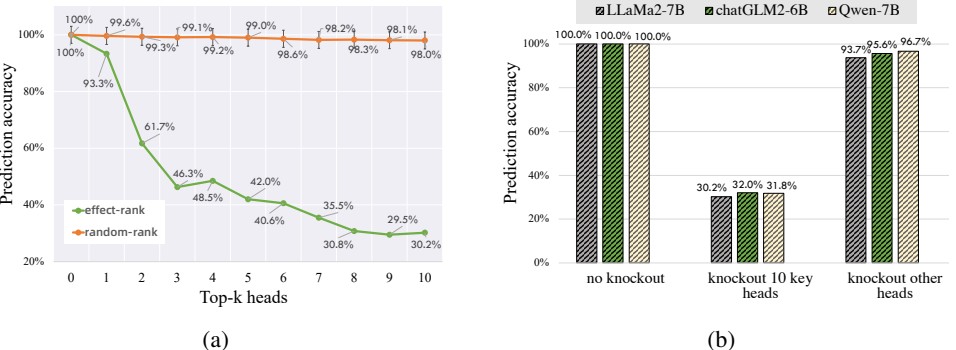

Figure 6: (a) The influence on prediction accuracy after knocking out top-k attention heads that are sorted by the effect of each head on logits ("effect-rank"), and knocking out randomly-sorted top-k heads ("random-rank"). (b) The performance comparison between knocking out 10 key heads and knocking out all heads except the 10 key heads.

and knocked out one by one. The green curve means attention heads are sorted by the effect of each head on logits, termed as "effect-rank". The orange curve means attention heads are sorted randomly, termed as "random-rank". As the heads are gradually knocked out, the performance of the model drops sharply in "effect-rank", while keeping stable (relatively minor effect within 2%) in "random-rank". More comparisons of another two models are included in Appendix D.

In Figure 6b, we directly remove all heads except the top-10 heads. Only knocking out the top-10 key heads leads to a significant performance decline ($-69.8\%$), while knocking out all other heads except the top-10 shows a relatively negligible performance decline ($-6.3\%$). Furthermore, we delve into the top-5 prediction probability and show several representative cases in Figure 7. After knocking out key heads, the model becomes largely confused to output incorrect numbers, while the original model can predict correctly with a much higher probability. The above results demonstrate that the discovered components play an especially important role in the language model's ability to complete the addition task. To our surprise, even when the majority of heads are knocked out (*e.g.*, leaving only the top-10 key heads of all 1024 heads in LLaMA2-7B), the model is still capable of performing the addition task.

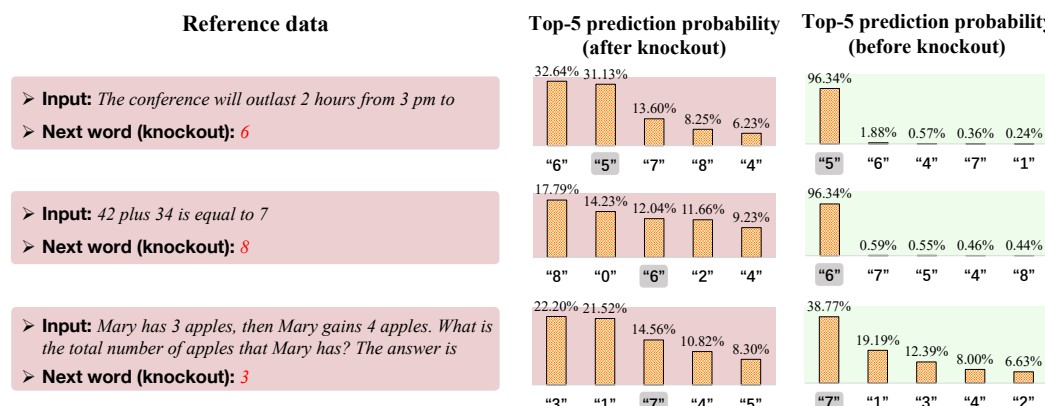

Figure 7: After knocking out the key heads, LLaMA2-7B provides incorrect predictions on reference data, which can be correctly answered by the original model with high prediction probabilities.

## 4.4 TRANSFER TO OTHER RELATED TASKS.

To verify that the behavior of the key heads identified is a general mechanism, instead of memorizing specific cases, we perform the knockout experiments on the unseen tasks of addition and subtraction.

| Experiment setting | Reference data | Top-5 prediction probability (after knockout) | Top-5 prediction probability (before knockout) |
|---|---|---|---|
| Transfer to unseen addition task | ➤ Input: *14 + 3 = 1* 
 ➤ Next word (knockout): *5* 

 ➤ Input: *The addition of 3 and 4 is* 
 ➤ Next word (knockout): *1* | | |
| Transfer to subtraction task | ➤ Input: *14 - 3 = 1* 
 ➤ Next word (knockout): *4* 

 ➤ Input: *From the year 1734 to the year 1736, the war lasted* 
 ➤ Next word (knockout): *1* 

 ➤ Input: *7 minus 3 is equal to* 
 ➤ Next word (knockout): *7* 

 ➤ Input: *Mary has 7 apples, then Mary loses 3 apples. What is the total number of apples that Mary has? The answer is* 
 ➤ Next word (knockout): *7* | | |

Figure 8: When testing on the reference data of unseen addition task and subtraction task, LLaMA2-7B provides incorrect predictions after knocking out key heads.

To begin with, we evaluate the model's prediction accuracy on a newly created addition dataset, which contains sentence templates that have not been encountered before (*e.g.*, the original addition equation "$\{A\} + \{B\} = \{C\}$"). When knocking out those discovered key heads, we notice a sharp decline in the model's performance from a perfect accuracy of $100\%$ to $35\%$. Additionally, in Figure 8 we observe a similar phenomenon in the prediction probabilities as in Figure 7.

Furthermore, since both subtraction and addition tasks require perception and computation of numbers in a similar manner, we speculate that the key heads may also contribute to solving subtraction problems. We test the model on a subtraction dataset with sentences built based on templates "$\{A\}$ - $\{B\} = \{C\}$". When the key heads are deactivated, the model observes a performance drop of $52\%$. By examining the attention patterns on subtraction sentences, these heads still attend particularly to the number tokens, as shown in Figure 9. This observation provides an explanation for why the de-

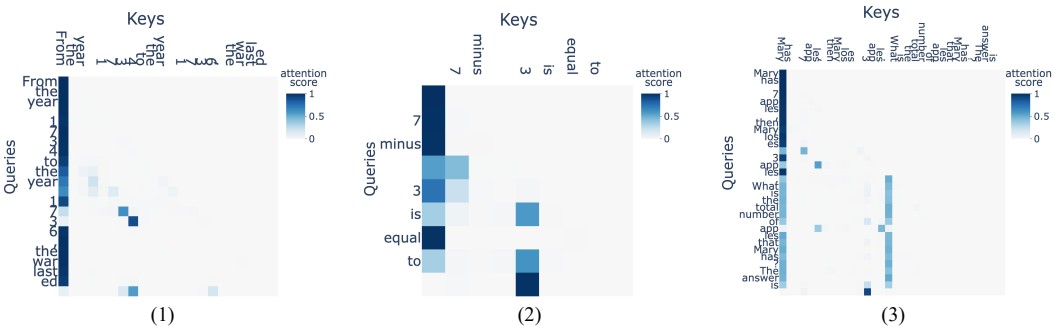

Figure 9: The attention patterns of the key heads (*e.g.*, 13.11 in LLaMA2-7B) on sentences with the subtraction logic.

activation of the key heads can influence the model's perception on number tokens and consequently affect its performance on the subtraction task.

The above results demonstrate that the functionality of key heads can be generalized to different tasks that need mathematical computation. More results of another two language models in Appendix D further validate this claim.

### 4.5  Discussion: Which heads affect the key heads?

Given that the key heads are primarily responsible for the output logits, we next ask what heads they depend on. As illustrated in Section 2, there are three ways to influence an attention head: by its Query (Q), Key (K), or Value (V). We again use path patching, to identify the heads that affect the Q/K/V vectors of the key heads.

For each head $h$, we patch a set of paths that originate from $h$ to the key heads' Q/K/V and measure the effect on the logit of token $\{\texttt{C}\}$. The experimental results in Figure 10 reveal that almost no head directly influences the key heads' keys and values, except for head $12.22$ which has a slight effect on the queries ($\pm 0.9\%$). In comparison to the logit changes ($\pm 10\%$) in Figure 3, we assume that the earlier-layer heads impose very small influence on the key heads. In other words, the key heads primarily operate on the input tokens only through residual streams or MLPs. Considering that the clearest path forward would require individually interpretable neurons in MLPs, further investigation of the effect of MLPs on the mathematical computation is left for future work.

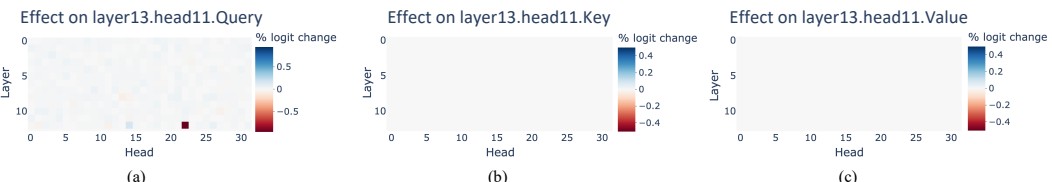

Figure 10: We search for each head $h$ directly affecting the Query, Key, and Value of attention head 13.11 in LLaMA2-7B, and measure the changes in the logit of the right answer.

### 5  Conclusion

In this study, we have identified, understood, and validated the crucial attention heads responsible for the mathematical addition capability of three LLMs. Our research delivers several interesting findings. 1) Only a few heads sparsely distributed in the middle layers contribute to the addition task significantly. 2) The identified key heads particularly attend to number tokens across various types of sentences. 3) Knocking out the key heads has a considerably larger impact on model performance compared to knocking out all remaining heads. All these findings contribute to a better understanding of the inner mechanism of LLMs. A more thorough study on the subtraction task as well as the validation on more computation tasks (*e.g.*, multiplication and division, etc.) is left for future work.

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

# 6  APPENDIX

## LIST OF REVISIONS

## A  DATASET TEMPLATES

We have included a comprehensive list of the templates used in this work as shown in Figure 11. Each name was randomly selected from a pool of 100 English first names, while the objects, verbs, and events were chosen from a curated list of 20 common words. For the datasets used in Section 4.4, we construct the sentences in different contexts with addition and subtraction logic, as shown in Figure 12 and Figure 13.

| |
|---|
| The `<EVENT>` `<VERB>` {A} years from the year `<YYY>`{B} to the year `<YYY>`{C} |
| The `<EVENT>` `<VERB>` {A} years from `<YYY>`{B} to `<YYY>`{C} |
| The `<EVENT>` `<VERB>` {A} days from `<MONTH>` {B} to `<MONTH>` {C} |
| The `<EVENT>` will `<VERB>` {A} days from `<MONTH>` {B} to `<MONTH>` {C} |
| The `<EVENT>` `<VERB>` {A} hours from {B} pm to {C} |
| The `<EVENT>` will `<VERB>` {A} hours from {B} pm to {C} |
| The `<EVENT>` `<VERB>` {A} hours from {B} am to {C} |
| The `<EVENT>` will `<VERB>` {A} hours from {B} am to {C} |
| {A} plus {B} equals to {C} |
| {A} plus {B} is equal to {C} |
| `<A1>`{A} plus `<B1>`{B} equals to `<C1>`{C} |
| `<A1>`{A} plus `<B1>`{B} is equal to `<C1>`{C} |
| {A} add {B} equals to {C} |
| {A} add {B} is equal to {C} |
| `<A1>`{A} add `<B1>`{B} equals to `<C1>`{C} |
| `<A1>`{A} add `<B1>`{B} is equal to `<C1>`{C} |
| `<NAME>` has {A} `<OBJECT>`, then `<NAME>` `<VERB>` {B} `<OBJECT>`. What's the total number of `<OBJECT>` that `<NAME>` has? The answer is {C} |
| `<NAME>` `<VERB>` {A} `<OBJECT>`, and `<NAME2>` `<VERB>` {B} `<OBJECT>`. What's the total number of `<OBJECT>` that they `<VERB>`? The answer is {C} |
| `<NAME>` has {A} `<OBJECT>`, and `<NAME2>` has {B} `<OBJECT>`. What's the total number of `<OBJECT>` that they have? The answer is {C} |
| `<NAME>` `<VERB>` {A} `<OBJECT>` yesterday, and `<NAME>` `<VERB>` {B} `<OBJECT>` today. What's the total number of `<OBJECT>` that `<NAME>` `<VERB>`? The answer is {C} |

Figure 11: Templates used in the addition dataset. All templates in the table involve the addition logic "{A} + {B} = {C}", but have different linguistic meanings like "time span", "number calculation", and "object accumulation".

| |
|---|
| {A} + {B} = {C} |
| `<A1>`{A} + `<B1>`{B} = `<C1>`{C} |
| The addition of {A} and {B} is {C} |
| The addition of `<A1>`{A} and `<B1>`{B} is `<C1>`{C} |
| The sum of {A} and {B} is {C} |
| The sum of `<A1>`{A} and `<B1>`{B} is `<C1>`{C} |

Figure 12: Templates used in the dataset when transferring to *unseen* addition task.

| |
|---|
| {A} - {B} = {C} |
| `<A1>`{A} - `<B1>`{B} = `<C1>`{C} |
| From the year `<YYY>`{B} to the year `<YYY>`{A} , the `<EVENT>` `<VERB>` {C} |
| From `<YYY>`{B} to `<YYY>`{A} , the `<EVENT>` `<VERB>` {C} |
| {A} minus {B} equals to {C} |
| {A} minus {B} is equal to {C} |
| `<NAME>` has {A} `<OBJECT>`, then `<NAME>` `<VERB>` {B} `<OBJECT>`. What's the total number of `<OBJECT>` that `<NAME>` has? The answer is {C} |
| `<NAME>` had {A} `<OBJECT>` yesterday, then `<NAME>` `<VERB>` {B} `<OBJECT>` today. What's the total number of `<OBJECT>` that `<NAME>` has? The answer is {C} |

Figure 13: Templates used in the dataset when transferring to subtraction task. All templates in the table imitate the addition templates but involve the subtraction logic "{A} - {B} = {C}".

## B    COMPARISON OF KEY HEADS ON FOUR MATHEMATICAL TASKS.

To investigate the distribution of key heads across four mathematical tasks (*i.e.*, addition, subtraction, multiplication, and division), we conduct the path patching experiments using the templates in Figure 14. The results shown in Figure 15 reveal that: (i) the sparsity of key heads remains consistent across all four tasks (less than $1.0\%$ of all heads). (ii) The key heads mainly distribute in the middle layers. The phenomena are analogous to the primary findings on the addition task (Section 4.1), demonstrating the potential of extending the observed effects of the addition task to other mathematical tasks.

We compare the location of key heads across four mathematical tasks. An interesting finding is that the key heads used in "subtraction" and "addition" tasks overlapped significantly, as did the key heads used in "multiplication" and "division" tasks. Moreover, the four tasks share the heads (*e.g.*, 13.11 and 12.22) that deliver the most significant effects, while they have task-specific heads that only emerge in its own task. These findings suggest that LLMs exhibit behavior aligned with human thinking to some extent, since "subtraction-addition" and "multiplication-division" are opposite mathematical operations.

| Addition | Subtraction |
|---|---|
| {A} + {B} = | {A} - {B} = |
| The sum of {A} and {B} is | The difference between {A} and {B} is |
| Q: What is {A} plus {B}? A: | Q: What is {A} minus {B}? A: |
| Q: How much is {A} plus {B}? A: | Q: How much is {A} minus {B}? A: |
| **Multiplication** | **Division** |
| {A} * {B} = | {A} / {B} = |
| The product of {A} and {B} is | The ratio between {A} and {B} is |
| Q: What is {A} times {B}? A: | Q: What is {A} over {B}? A: |
| Q: How much is {A} times {B}? A: | Q: How much is {A} over {B}? A: |

Figure 14: Templates used to investigate the mathematical tasks of addition, subtraction, multiplication, and division. "Q" and "A" are the abbreviation for "Question" and "Answer", respectively.

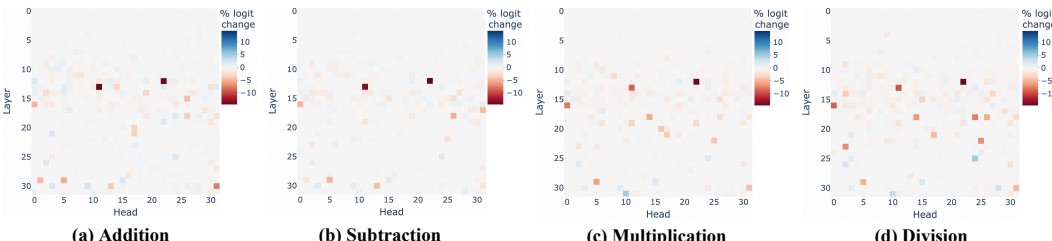

(a) Addition         (b) Subtraction         (c) Multiplication         (d) Division

Figure 15: We conduct path patching experiments on LLaMA2-7B across four mathematical tasks, by searching for each head $h$ directly affecting the logit of the right answer.

Interestingly, when examining subtraction and addition tasks, we could summarize two insightful symmetries between them. (i) The identified key heads of two tasks are almost the same, albeit with different magnitude of the effect. This phenomenon could reveal the symmetry of key head "*location*" in addition and subtraction. (ii) These heads particularly attend to the number tokens regardless of whether they are given addition or subtraction sentences (shown in Section 4.4). This phenomenon could reveal the symmetry of key head "*behavior*" in addition and subtraction.

## C    ATTENTION PATTERNS IN ANOTHER TWO LANGUAGE MODELS.

In Figure 16 and Figure 17, we list the attention patterns of the identified key heads (*e.g.*, 19.15 in chatGLM2-6B, 18.4 in Qwen-7B) on different samples. Whether the sentences containing the addition logic (samples 1-3), the subtraction logic (samples 4-6), or even sentences without linguistic

meanings (samples 7-9), the key heads have particularly higher attention scores on number tokens in the sentences. These results further validate the claimed functionality of these key heads.

Figure 16: The attention patterns of head 18.4 in Qwen-7B on different samples. The samples (1-3) are sentences with the addition logic. The samples (4-6) are sentences with the subtraction logic. The samples (7-9) are randomly constructed sentences with no linguistic meaning.

# D  KNOCKOUT RESULTS OF ANOTHER TWO LANGUAGE MODELS.

In Figure 18, we measure the prediction accuracy after knocking out the identified key heads in different language models. As we increase the number of knocked key heads, the model's performance experiences a significant decline followed by a gradual stabilization. This pattern is consistent across three different models, providing further evidence that the identified key heads play a crucial role in completing the addition task.

# E  COMPARISON OF KEY HEADS IN DIFFERENT LLMS.

In Figure 3, we present the identified key heads in different LLMs of LLaMA2-7B, Qwen-7B and chatGLM2-6B. Despite the sparsity of key heads compared to the total number of heads (*e.g.*, 1024 in LLaMA2-7B and Qwen-7B, 896 in chatGLM2-6B), LLaMA2 exhibits a relatively denser distribution of results compared to the other two models. Apart from the key heads 13.11 and 12.22,

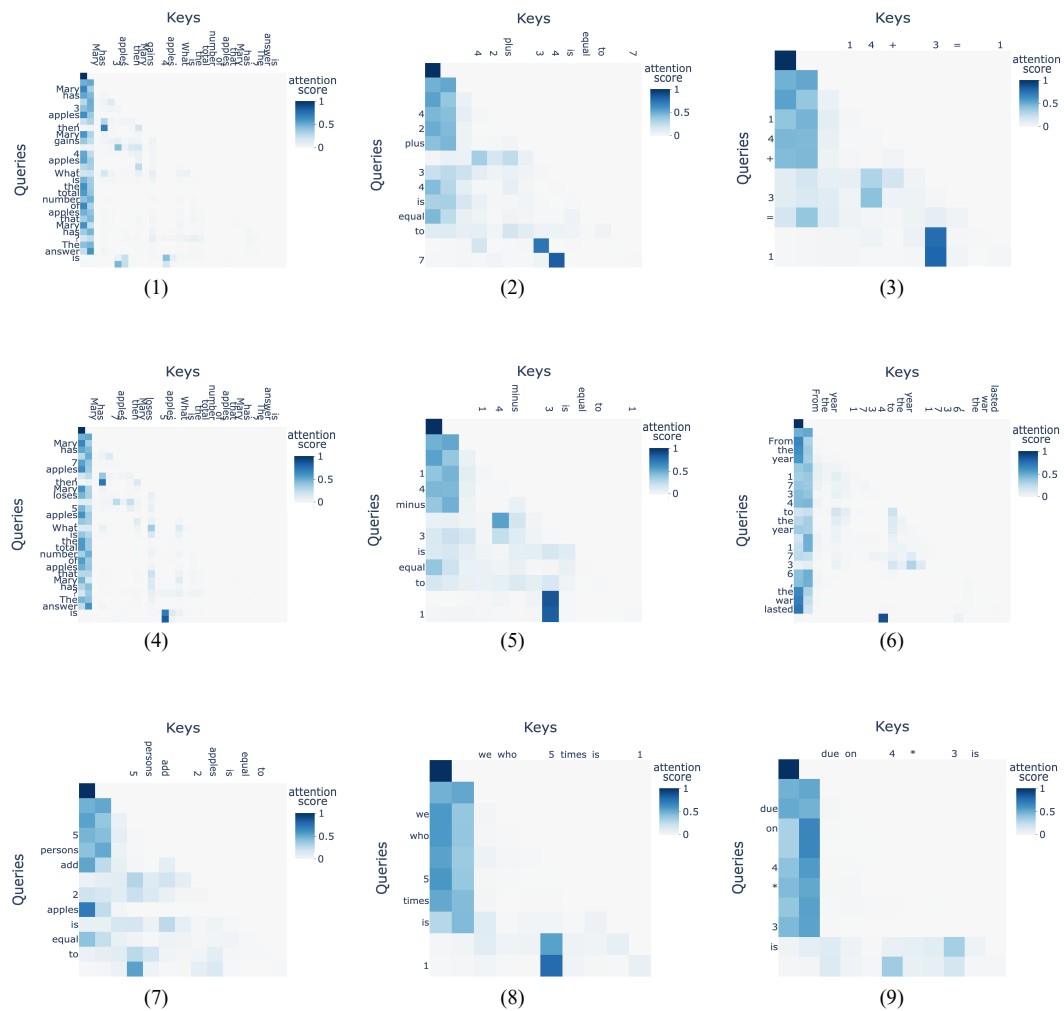

Figure 17: The attention patterns of head 19.15 in chatGLM2-6B on different samples. The samples (1-3) are sentences with the addition logic. The samples (4-6) are sentences with the subtraction logic. The samples (7-9) are randomly constructed sentences with no linguistic meaning.

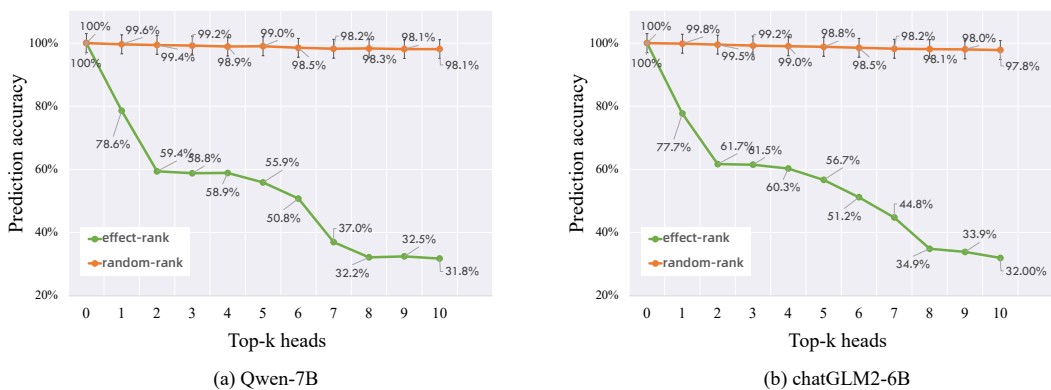

Figure 18: The influence on prediction accuracy after knocking out top-k attention heads in the language model of Qwen-7B and chatGLM2-6B. The heads are sorted by the effect of each head on logits ("effect-rank"), or randomly sorted ("random-rank").

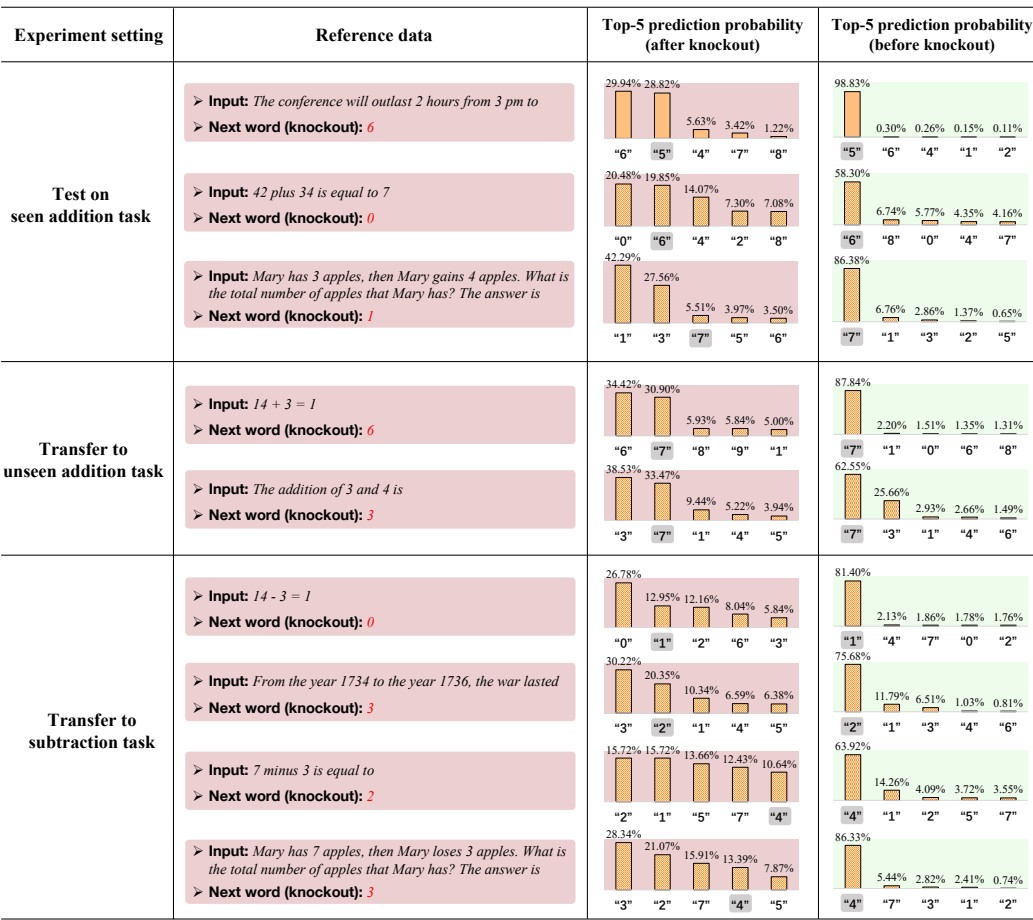

Figure 19: When testing on the reference data of seen addition tasks, unseen addition tasks and subtraction tasks, Qwen-7B provides incorrect predictions after knocking out key heads.

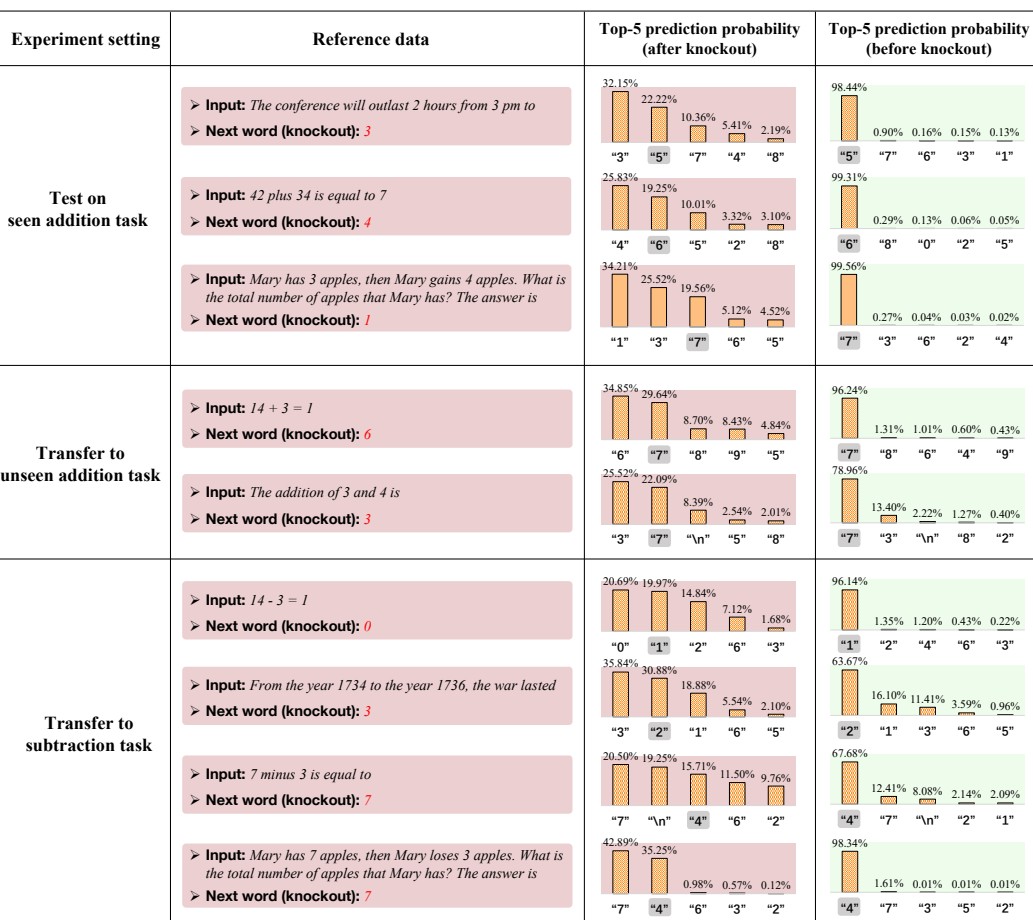

Figure 20: When testing on the reference data of seen addition tasks, unseen addition tasks, and subtraction tasks, chatGLM2-6B provides incorrect predictions after knocking out key heads.

other heads such as 16.0, 14.19, and 15.15 also make a difference on the output logits. It indicates that LLaMA2-7B involves more components in completing the addition task. We hypothesize that this is because LLaMA2-7B demonstrates superior comprehension abilities for mathematical tasks, resulting in more obvious response patterns. To assess the model's proficiency in understanding mathematical tasks, we conducted experiments to evaluate whether the model comprehends the aims of the mathematical task by generating higher logits for numerical tokens. Based on the reference data $X_\mathrm{r}$, we compute the average prediction probability $P_{avg}$ of numerical tokens 1-9 for different models (LLaMA2-7B: 99.26% vs. Qwen-7B: 97.55% vs. chatGLM2-6B: 94.68%). The higher $P_{avg}$ of LLaMA2-7B shows that it generates number tokens with a greater confidence level, thus demonstrating a better understanding of the mathematical task. This finding provides an explanation of why LLaMA2-7B involves more key heads. Further investigation is required for a more comprehensive analysis.

