# OpenReview forum: "Interpreting the Inner Mechanisms of Large Language Models in Mathematical Addition"
_ICLR.cc/2024/Conference — Submitted to ICLR 2024_

### Official Review · Reviewer_h6js · 2023-10-21

**Soundness:** 3 good
**Presentation:** 3 good
**Contribution:** 2 fair
**Rating:** 3
**Confidence:** 4

**Summary:**

The paper identifies attention heads which play a key role for addition and subtraction in transformer-based language models. The authors identify several such heads in three different models and demonstrate that removing them destroys the ability of the model to perform arithmetic.

**Strengths:**

1. The analysis is thorough and rigorous.
2. The paper is clearly written, and the presentation is well-organised and presented.
3. On top of identifying the “key heads” being focused on addition, the paper shows that the same heads are also involved in subtraction. While this might be intuitive, considering that one of these tasks is the opposite of the other, it is not obvious that an LLM would discover and utilize this duality. However, it is unclear whether that is because the heads focus only on numbers or because they are utilising the duality of summation and subtraction.
4. The paper recognises that later heads depend on earlier ones and attempts to analyse these dependencies (although it appears there are none).

**Weaknesses:**

1. The paper identifies attention heads that take part in the processing of summation but does not look into or explain what each of the “key heads” actually does and what is the mechanism through which it contributes to summation. Therefore, the paper focuses on _localization_ of the heads that partake in summation, rather than _interpreting_ them.

2. The paper does not look at alternative representations of numbers. For example, in words (“two” instead of 2), Roman numerals (II instead of 2), and other languages (二 or ٢ instead of 2). The lack of such analysis leaves the question open whether these heads simply attend to numerical tokens or whether they are involved in higher-order reasoning about numbers and arithmetic.

3. Related to the above, the paper seems to focus only on single-digit summation. It is unclear whether the results would translate to the summation of larger numbers (or more than two numbers). This is important as prior works have shown that the ability of LLMs to do arithmetic quickly decreases with the increase of the number of digits. It would be interesting to see if your analysis would be able to provide insights into this phenomenon.

4. I am not sure how to read the attention patterns in Fig. 4. How can the attention be negative? In fact, it does not seem that these heads attend to all numbers. The first head seems to attend to the completion of “or” with “anges” and the full stop. Both heads seem to attend only to 3 while solving the task would also require attention to 5. Therefore, it is not clear how these heads participate in performing summation.

5. The paper looks predominantly at attention heads. However, it is well known that a lot of the computation and processing happens in the MLPs. Hence, a full picture of the interoperation of the mechanisms for summation should also include the MLPs

**Questions:**

1. Does knocking out the heads have effects on other tasks, i.e. are these heads only important for arithmetic or are they polysemantic?

2. In the Introduction, you say _“Contrary to expectations that LLMs may involve diverse attention heads across all layers, our findings reveal that only a small percentage (0.5%) of the attention heads significantly impact the model’s performance.”_ However, this is exactly the expectation: attention heads have diverse functions so it is not surprising that only a few of them would be involved in summation.

3. In the Introduction, you say _“Remarkably, only the absence of these heads leads to a notable decline in model performance.”_ But this can’t be true. Surely there are many other weights that, if perturbed, would result in a significant decline in model performance (e.g. the embedding matrix or the final output projection matrix).

4. In Section 4.2, how do you decouple the effect of the individual heads? In the implementation of LLAMA there are no separate $W_O$ for each head but a single one that is applied to the concatenation of all the heads. Therefore, it mixes information across heads. How do you resolve this?

5. In Figure 4 left, how do you know that the effect you see is because of the heads specialising in numbers and not because your test sequences have numbers in them? I’d be curious to see how this plot and the rest of your analysis would look like if applied to sentences which have nothing to do with numbers and arithmetic. Possibly the heads that you have found to be important for arithmetic would be especially unimportant for other tasks.

---

> ### Author Response · Authors · 2023-11-16
> **Response to Reviewer h6js [part 1/2]**
>
> We would like to thank the reviewer for taking the time to review our work. We are grateful that you find our research to be thorough, rigorous, well-organised and presented. Thank you for acknowledging our efforts to the findings of key heads in addition. According to your valuable comments, we provide detailed feedback.
>
> **Q1**: More interpretation for attention heads and MLPs:
> > (a) "the paper focuses on localization of the heads that partake in summation, rather than interpreting them"
> > (b) "a full picture of the interoperation of the mechanisms for summation should also include the MLPs"
>
> **Ans for Q1:**
> **(a)** Thank you for bringing up the insightful comment!
> * We agree that delving into the *"causal mechanism"* of addition calculation is a quite promising direction. In this study, we follow the recent research in mechanistic interpretability [1, 2] that adapt the theory of causal mediation analysis to measure the *"causal effect"* of component mediators to the model behavior. We would like to highlight the identification of key heads *implicitly* interpret the model behavior (e.g. addition calculation).
> * Before this study, there was a lack of clear interpretation regarding several important questions of addition calculation in LLMs.
>   (i) Are there attention heads consistently involved in addition calculation?
>   (ii) If yes, are they sparsely or densely distributed?
>   (iii) In the case of sparsity, where exactly are these key heads located?
> To shed light on these matters, we performed sufficient experiments using different models and data formats, striving to present the observed phenomena in a clear and understandable manner.
>
> We hope that our efforts could contribute to making the inner workings of LLMs in addition more understandable to humans.
>
> **(b)** Thank you for the valuable suggestion on MLPs.
> * We added one experiment of path patching to measure the causal effect of each MLP layer. The results indicate that: (i) the early MLP layers 0-13 before the key heads (e.g., 13.11) have a slight effect on the output (approximately $\pm$0.5%). (ii) The late MLP layers 14-31 after the key heads exhibit a much larger effect (approximately $\pm$10.0%). These findings may reveal that the key heads are responsible for attending to number tokens, while the following MLP layers process the number tokens.
> * We also noticed that previous works [1, 2] tried to explain MLPs for performing computations and retrieving facts. However, it necessitates thorough and rigorous analysis to make the detailed process in MLPs human-understandable, and interpret the collaboration between attention heads and MLPs. We believe that making the calculation fully understandable to humans is an intriguing direction that warrants sustained research efforts.
>
>
> [1] Locating and Editing Factual Associations in GPT. In NeurIPS 2022.
> [2] Transformer Feed-Forward Layers Are Key-Value Memories. In EMNLP 2021.
>
> **Q2**: More alternatives of numbers:
> > (a) "The paper does not look at alternative representations of numbers. For example, in words (“two” instead of 2), Roman numerals (II instead of 2), and other languages (二 or ٢ instead of 2)."
> > (b) "It is unclear whether the results would translate to the summation of larger numbers (or more than two numbers)"
>
> **Ans for Q2:**
> Thanks for your constructive suggestion. We conduct the knockout experiments on the following number formats. The table shows one representative sample for each format and the accuracy change after knocking out the key heads in LLaMA2-7B.
>
> | settings | formats | samples | accuracy |
> |-------| ------- | ------- | ------- |
> |    a1     | English  | "The addition of two and five is " |   -45%      |
> |    a2   |  Chinese |     "两和五相加的结果是"   |    -5%     |
> |    a3     |  Roman |   "II + V = "  |   -5%      |
> |     b1    | two-digit  |   "35 + 42 = " |    -69%     |
> |     b2   | three-digit  |   "154 + 243 = " |     -72%    |
> |     b3  | four-digit  |   "1524 + 3463 = " |    -76%     |
>
> **(a)** To investigate the effects of numbers represented in different languages, we conduct the experiments as settings a1-a3. After knocking out the key heads identified based on Arabic numbers, the prediction accuracy for samples containing English numbers still decreases noticeably, while there is almost no decline in accuracy for samples containing Chinese and Roman numbers. We assume this is because the training corpus of LLaMA2 mainly consists of English language texts, which could explain the disparity in performance.

---

> > ### Author Response · Authors · 2023-11-16
> > **Response to Reviewer h6js [part 2/2]**
> >
> > **(b)** In response to the issue of one-digit addition, we conduct the experiments as settings b1-b3.
> > * After knocking out the key heads identified based on one-digit addition, the performance for samples containing multi-digit numbers decreases significantly. When scaling to larger numbers, the performance drops more greatly. We assume this is because one-digit addition serves as the fundamental computation *"unit"* for multi-digit addition. The perturbation effects on one-digit addition could be accumulated when scaling to larger numbers.
> > * Moreover, the reason why we experiment on one-digit addition in the primary experiments is that the adopted three LLMs (e.g., LLaMA2-7B) tokenize each digit individually (e.g., '42' is tokenized to '4' and '2'). We follow the one-digit nature of LLMs for simplicity in generating large-scale data.
> >
> > The above studies may provide an explanation for why *"the ability of LLMs to do arithmetic quickly decreases with the increase of the number of digits"*.
> >
> > **Q3**: More illustration:
> > > "I am not sure how to read the attention patterns in Fig. 4. How can the attention be negative?"
> >
> > **Ans for Q3:**
> > Thanks for your careful review. This is a visualization typo. The attention is bounded in [0, 1]. As for the selected example in Fig. 4, we find that it particularly attends to the token "3" by Head 13.11 and Head 12.22, while Head 17.31 attends to the tokens "5" and "3" simultaneously. We have revised the potentially confusing issue in the revision, by replacing it with another representative example.
> >
> > **Q4**: More evaluation:
> > > "Does knocking out the heads have effects on other tasks, i.e. are these heads only important for arithmetic or are they polysemantic?"
> >
> > **Ans for Q4:**
> > Thanks for pointing out this insightful issue. We conduct the knockout experiment on the dataset CSQA [3] to evaluate the Chain-of-Thought (CoT) task. This dataset focuses on answering commonsense questions that require prior knowledge and multi-step reasoning. We find that knocking out the key heads in addition task has almost no effect on the CoT task, suggesting a certain level of monosemanticity on this dataset. Moreover, we notice that recent research [4] provides a detailed analysis of the monosemanticity and polysemanticity of network neurons.
> >
> > [3] Commonsenseqa: A question answering challenge targeting commonsense knowledge. In ACL 2019.
> > [4] Towards Monosemanticity: Decomposing Language Models With Dictionary Learning. In Transformer Circuits Thread 2023.
> >
> > **Q5**: Writing issues:
> > > (a) "In the Introduction, you say 'Contrary to expectations that ...'. However, this is exactly the expectation ..."
> > > (b) "In the Introduction, you say 'Remarkably, only the absence of these heads ...'. But this can’t be true ..."
> >
> > **Ans for Q5:**
> > Thanks for your careful review. We have revised the representations following the suggestion.
> > **(a)** We remove *'Contrary to expectations that LLMs may involve diverse attention heads across all layers,'* for more precise expression.
> > **(b)** We change the original sentence to *'Remarkably, the absence of these heads leads to a relatively notable decline in model performance, compared to the absence of randomly-selected heads.'*.
> >
> > The above changes have been highlighted in blue in our revision.
> >
> > **Q6**: Implementation detail:
> > > "In Section 4.2, how do you decouple the effect of the individual heads?"
> >
> > **Ans for Q6:**
> > Thanks for pointing out this issue. We follow the implementation in [5] to split $W_O$ into equal size blocks for each head $\left [  W_O^{h_1}, W_O^{h_2}, \cdots \right ]$. As is verified in [6], it's equivalent to running heads independently, multiplying each by its own output matrix, and adding them into the residual stream.
> >
> > [5] Interpretability in the Wild: a Circuit for Indirect Object Identification in GPT-2 small. In ICLR 2023.
> > [6] A Mathematical Framework for Transformer Circuits. In Transformer Circuits Thread 2021.
> >
> > **Q7**: More analysis:
> > > "In Figure 4 left, how do you know that the effect you see is because of the heads specialising in numbers and not because your test sequences have numbers in them?"
> >
> > **Ans for Q7:**
> > Thank you for bringing up this concern. We conduct the plot experiment based on the counterfactual sentences. These cases contain no arithmetic, i.e., excluding addition logic, and implicitly have nothing to do with numbers. The findings indicate that the key heads still exhibit a relatively lower attention probability towards non-numeric tokens, similar to the observation depicted in Figure 4. This phenomenon supports the claimed effects of the key heads at a certain level.

---

> ### Author Response · Authors · 2023-11-20
> **Welcome for more discussions (#h6js)**
>
> Thanks for your valuable time in reviewing and constructive comments, according to which we have tried our best to answer the questions and carefully revise the paper. Here is a **summary of our response** for your convenience:
>
> - (1) **Interpretation issues**: Following your constructive comments, we have provided more studies to interpret the behavior of LLMs w.r.t. key heads and MLPs. We also conducted experiments on MLPs to provide a full picture of the inner workings of LLMs in addition.
>
> - (2) **Experiment issues**: Following your valuable suggestions, we have conducted experiments on more number representations, including different languages and multi-digit numbers. The results of scaling to larger numbers provide an insightful explanation for "the ability of LLMs to do arithmetic quickly decreases with the increase of the number of digits". Moreover, we also added experiments to evaluate the polysemanticity of key heads, and delve deeper into the effects of numbers and arithmetic.
>
> - (3) **Writing issues**: Following your kind suggestions,  we have clarified the potentially confusing issues, including the visualization case in Fig. 4, the expression in Introduction, and implementation details about $W_O$. We have also revised these issues, highlighted in blue, in our revision.
>
> We humbly hope our repsonse has addressed your concerns. If you have any additional concerns or comments that we may have missed in our responses, we would be most grateful for any further feedback from you to help us further enhance our work.
>
>
> Best regards
>
> Authors of #3226

---

> ### Author Response · Authors · 2023-11-21
> **Window for responsing and draft updating is closing**
>
> Dear Reviewer #h6js,
>
> Thanks very much for your time and valuable comments. We understand you're busy. But as the window for responsing and paper revision is closing, would you mind checking our response ([a brief summary](https://openreview.net/forum?id=VpCqrMMGVm&noteId=XmiOLrJmzA), and [details](https://openreview.net/forum?id=VpCqrMMGVm&noteId=g2v0fTbFQE)) and confirm whether you have any further questions? We are very glad to provide answers and revision to your further questions.
>
> Best regards and thanks,
>
> Authors of #3226

---

> > ### Comment · Reviewer_h6js · 2023-11-22
> >
> > Thank you for the detailed response.
> >
> > > We would like to highlight the identification of key heads _implicitly_ interpret the model behavior (e.g. addition calculation).
> >
> > Considering that _interpretability_ already is a vague term that you have not formally defined, I am concerned that _implicit interpretability_ has even less practical or scientific value.
> >
> > > Before this study, there was a lack of clear interpretation regarding several important questions of addition calculation in LLMs. (i) Are there attention heads consistently involved in addition calculation? (ii) If yes, are they sparsely or densely distributed? (iii) In the case of sparsity, where exactly are these key heads located?
> > To shed light on these matters, we performed sufficient experiments using different models and data formats, striving to present the observed phenomena in a clear and understandable manner.
> >
> > I agree that these questions may not have been studied before. But it is not clear to me why are these questions interesting, important, or relevant. Why is addition so important? (Much more than the myriad of other operations and behaviors that one could study?) What value does it bring knowing where these heads are located?
> >
> > > We added one experiment of path patching to measure the causal effect of each MLP layer. The results indicate that: (i) the early MLP layers 0-13 before the key heads (e.g., 13.11) have a slight effect on the output (approximately
> > 0.5%). (ii) The late MLP layers 14-31 after the key heads exhibit a much larger effect (approximately
> > 10.0%). These findings may reveal that the key heads are responsible for attending to number tokens, while the following MLP layers process the number tokens.
> >
> > Your response seem to confirm my concerns. The MLPs are likely involved in the computation and you cannot ignore them. Moreover, the Path Patching procedure you describe maintains the connections through the MLPs. Therefore, how can we disambiguate which components are "responsible" when they might be _jointly necessary conditions_ to exhibit the behavior you study.
> >
> > > We also noticed that previous works [1, 2] tried to explain MLPs for performing computations and retrieving facts. However, it necessitates thorough and rigorous analysis to make the detailed process in MLPs human-understandable, and interpret the collaboration between attention heads and MLPs. We believe that making the calculation fully understandable to humans is an intriguing direction that warrants sustained research efforts.
> >
> > Your abstract claims that you _"take the first attempt to reveal a specific mechanism relating to how LLMs implement the reasoning task of a mathematical addition, e.g., scenarios involving simple one-digit integer addition"_. That expects that you would have included this _"thorough and rigorous analysis"_...
> >
> > > Thanks for your constructive suggestion. We conduct the knockout experiments on the following number formats. The table shows one representative sample for each format and the accuracy change after knocking out the key heads in LLaMA2-7B.
> >
> > Thank you for running these additional experiments! They indeed confirm my concerns: If you managed to locate all the key heads involved in summation, then knocking them out would have also resulted in significantly lower accuracy for the Chinese and Roman cases. The fact that they see only 5% drop in performance shows that there are summation-relevant heads that your analysis has failed to identify.
> >
> > > Q3: More illustration
> >
> > The second example in Fig4 still doesn't exhibit the behavior you claim: the space after "equal to" should attend to "1" and "3", not to "2". Similarly, Fig 9.2 doesn’t attend to "7". From your response it also seems that these examples might have been cherry-picked to represent the behavior you want to demonstrate.
> >
> > Overall it feels like this paper applies existing mechanistic interpretability techniques to the problem of summing numbers. In light of the outstanding concerns above and (my subjective view) that this work does not contribute new knowledge and sufficient value to the community, I will lower my score.

---

> > > ### Author Response · Authors · 2023-11-23
> > > **Response to Reviewer h6js [part 1/2]**
> > >
> > > Thank you for taking time to post the detailed feedback before the window closes. We understand that your **subjective** goal is to contribute to our community through detailed comments.  Time is running out for discussion, so we will do our best to effectively clarify, as a sign of respect for your time and goals.
> > >
> > >
> > > > **Q1**: Considering that interpretability already is a vague term that you have not formally defined, I am concerned that implicit interpretability has even less practical or scientific value.
> > >
> > > **Ans for Q1**:
> > > We agree that interpretability is not mathematically defined in our paper. However, we believe whether the interpretability is a vague term is **subjective**. For instance, many outstanding efforts have been devoted or are devoting to the field of interpretability with clear definition [1,2,3,4].
> > >
> > > - As the cited work in our paper (cf. Sec. 3 Method), where interpretability has a non-mathematically definition. Following your valuable comments, we will involve the definition into our work.
> > >
> > > - We would like to note that, in our submission, we **did not** claim that we have completed the research on the implicit interpretability. The motivation is a possible proposal for your valuable comments regarding the interpretability for MLPs.
> > >
> > > - We mainly focus on the interpretability of attention heads, like previous outstanding works [3,4] on neurons, MLPs, etc. Thus, we overlook the neurons and MLPs.
> > >
> > >
> > > [1] Interpretability in the Wild: a Circuit for Indirect Object Identification in GPT-2 small. In ICLR 2023.
> > > [2] A mathematical framework for transformer circuits. In Transformer Circuits Thread 2021.
> > > [3] Locating and Editing Factual Associations in GPT. In NeurIPS 2022.
> > > [4] Transformer Feed-Forward Layers Are Key-Value Memories. In EMNLP 2021.
> > >
> > >
> > > > **Q2**: I agree that these questions may not have been studied before. But it is not clear to me why are these questions interesting, important, or relevant. Why is addition so important? (Much more than the myriad of other operations and behaviors that one could study?) What value does it bring knowing where these heads are located?
> > >
> > > **Ans for Q2**:
> > > We agree that what is interesting is not defined in our paper. However, we believe whether a study is interesting is **subjective**. In response to your valuable comments, please let us list our subjective interests.
> > >
> > > - Interpretability is crucial, especially as LLMs have rapidly developed in recent years. It is necessary to interpret the capabilities of LLMs.
> > > - However, we cannot study the interpretability of all tasks at the same time. Therefore, our focus in this paper is calculation as it is the basis for LLMs. Accordingly, we focus on one of the most fundamental ability in mathematical tasks: "addition".
> > > - In this work, we discover, understand, and validate the key heads involved in addition calculations in LLMs (not just the discovery step). We also find the key heads are consistently involved in addition and subtraction, which share the same spirit with human behavior.
> > >
> > >
> > > > **Q3**: Your response seem to confirm my concerns. The MLPs are likely involved in the computation and you cannot ignore them. Moreover, the Path Patching procedure you describe maintains the connections through the MLPs. Therefore, how can we disambiguate which components are "responsible" when they might be jointly necessary conditions to exhibit the behavior you study.
> > >
> > > **Ans for Q3**:
> > > We agree with your point that many components in a neural network are involved in the computation. However, we believe whether a component should be involved is **subjective**.
> > >
> > > - The bias term and LayerNormalization operation are also involved in the computation, thus, we are not prone to ingnore them. However, it is challenging to define the causal effects of these components when interpreting LLMs. Thus, previous outstanding works mainly focus on neurons, MLPs, and attention heads, overlooking other components.
> > >
> > > - We agree with that all components are jointly necessary conditions to exhibit the behavior. However, we would like to highlight the difference bwteen endogenous variables and causal mechanism. Specifically, we follow previous outstanding work to study the causal effect between endogenous variables and do not focus on the hard or soft causal intervention on causal mechanisms. We will highlight the diffenrence in our revision, due to limited time left for revision.
> > >
> > >
> > >
> > > > **Q4**: Your abstract claims that you "take the first attempt to reveal a specific mechanism relating to how LLMs implement the reasoning task of a mathematical addition, e.g., scenarios involving simple one-digit integer addition". That expects that you would have included this "thorough and rigorous analysis"...
> > >
> > > **Ans for Q4**:
> > > Thanks for your kind suggestion.

---

> > > > ### Author Response · Authors · 2023-11-23
> > > > **Response to Reviewer h6js [part 2/2]**
> > > >
> > > > > **Q5**: Thank you for running these additional experiments! They indeed confirm my concerns: If you managed to locate all the key heads involved in summation, then knocking them out would have also resulted in significantly lower accuracy for the Chinese and Roman cases. The fact that they see only 5% drop in performance shows that there are summation-relevant heads that your analysis has failed to identify.
> > > >
> > > > **Ans for Q5**:
> > > > We conducted these experiments following your valuable comments, which significantly contributed to our work.
> > > >
> > > > - We apologize for the misunderstanding. We claim that our method can identify key heads involved in summation, but it is challenging to show that our method can identify all attention heads involved in summation.
> > > > - We identify key heads because summation may exhibit various contexts, making all heads potentially critical.
> > > > - The key impact of these identified heads is demonstrated in our experiments. Specifically, the effect of knocking out these identified heads is significantly more significant than knocking out the same number of randomly selected heads.
> > > >
> > > >
> > > > > **Q6**: The second example in Fig4 still doesn't exhibit the behavior you claim: the space after "equal to" should attend to "1" and "3", not to "2". Similarly, Fig 9.2 doesn’t attend to "7". From your response it also seems that these examples might have been cherry-picked to represent the behavior you want to demonstrate.
> > > >
> > > > **Ans for Q6**:
> > > > We apologize for the misunderstanding. We did not claim that each key head attends to all number tokens. Instead, we claim that "these heads prioritize attending to the number tokens compared to the other tokens (i.e., not a number)". In Figure 4, we conducted both quantitative and qualitative studies, where we show the visualization cases in order to make the working mechanism of the discovered key heads more understandable.
> > > >
> > > >
> > > > We appreciate your time to review. We believe your valuable comments significantly promote the quality of our work. Thus, we've been looking forward to more of your valuable feedback over the past week, i.e., from 16 Nov. 2023 to 22 Nov. 2023. However, our response to your comments decreases your score from 6 to 3. Our response may cause some misunderstanding, and thus, we provide more explanations and clarifications before the end of discussion, i.e., **22 Nov. 2023 (AoE)**. We would appreciate it if you could take a look at our response.

---

> > > > > ### Comment · Reviewer_h6js · 2023-12-01
> > > > >
> > > > > Thank you so much for the detailed response. I appreciate that lowering my score must be frustrating and that I should have raised some of these issues when writing the original reviews. I apologize for that.
> > > > >
> > > > > > However, we believe whether the interpretability is a vague term is __subjective__. For instance, many outstanding efforts have been devoted or are devoting to the field of interpretability with clear definition [1,2,3,4].
> > > > >
> > > > > I am afraid I cannot find a formal definition of _interpretability_ in any of the four works you cite.
> > > > >
> > > > > > Interpretability is crucial, especially as LLMs have rapidly developed in recent years. It is necessary to interpret the capabilities of LLMs.
> > > > >     However, we cannot study the interpretability of all tasks at the same time. Therefore, our focus in this paper is calculation as it is the basis for LLMs. Accordingly, we focus on one of the most fundamental ability in mathematical tasks: "addition".
> > > > >     In this work, we discover, understand, and validate the key heads involved in addition calculations in LLMs (not just the discovery step). We also find the key heads are consistently involved in addition and subtraction, which share the same spirit with human behavior.
> > > > >
> > > > > I am still not convinced in the importance of localizing some of the heads participating in addition.
> > > > >
> > > > > My understanding of your contributions from your response is that you apply existing techniques to identify some attention heads of pre-trained transformers that attend to some number tokens and participate in summation when expressed in particular ways. Based on this, I will maintain my revised score.

---

### Official Review · Reviewer_7p7z · 2023-10-30

**Soundness:** 3 good
**Presentation:** 3 good
**Contribution:** 2 fair
**Rating:** 6
**Confidence:** 4

**Summary:**

In this study, the authors aim to delve into the underlying mechanisms of Large Language Models (LLMs) by analyzing attention heads at various layers in tasks that require the addition of two integers. Specifically, they focus on the LLAMA2-7B, Qwen-7B, and ChatGLM2-6B language models. Their findings reveal that a limited number of attention heads significantly influence the model's output, and these conclusions are drawn from a range of experiments. Furthermore, the authors show some preliminary results indicating that these same attention heads play a significant role in the performance of subtraction tasks.

**Strengths:**

Authors are tackling an important problem by aiming to understand the inner workings of LLMs. With the increased pace of advancements happening in the field, it is imperative to gain this understanding.

Authors tackle the problem in a clear manner, by coming up with a clean task (involving addition of 2 integers) and testing their hypothesis systematically.

Their findings indicate that a limited number of attention heads suffice for achieving strong performance across a range of addition tasks. Importantly, the methodology they introduce can prove valuable for conducting sensitivity analyses in other areas of interest and even facilitate model sparsification.

They validate their hypothesis on several LLMs and a few addition tasks. Additionally, their preliminary investigations reveal that the attention heads vital for addition tasks also exert a substantial influence on subtraction.

**Weaknesses:**

While the authors have indeed posed a clear problem and approached it systematically, I find the setup to be somewhat restrictive.

- Although the authors make a great effort to tackle the task of addition, their focus remains solely on the addition of two integers. It would be intriguing to see whether their findings extend to addition of multiple integers and rational numbers, as well as their applicability to problems involving multiple addition operations.

- The robustness of this study could be significantly enhanced if the authors were to conduct analogous experiments on subtraction, multiplication, and division. Such investigations would shed light on whether a select group of attention heads can consistently influence performance across all four mathematical operations.

**Questions:**

Please refer to weakness section. It would be great if authors have any additional insights regarding the points in weakness section.

---

> ### Author Response · Authors · 2023-11-16
> **Response to Reviewer 7p7z [part 1/2]**
>
> We sincerely thank the reviewer for taking the time to review our work. We appreciate that you find our work tackles an important problem in a clear and systematical manner, and is transferable to facilitates other areas. Thank you for acknowledging our efforts to the significant findings. According to your valuable comments, we provide detailed feedback.
>
> **Q1**: More experiments on addition task:
> > "their focus remains solely on the addition of two integers ... extend to addition of multiple integers and rational numbers, as well as their applicability to problems involving multiple addition operations."
>
> **Ans for Q1:**
> Thank you for the constructive suggestion.
> * We agree that it's necessary to take experiments on more formats of numbers besides one-digit integer numbers. The reason why we perform on one-digit addition in the primary experiments is that the adopted three LLMs (e.g., LLaMA2-7B) tokenize each digit individually (e.g., '42' is tokenized to '4' and '2'). We follow the one-digit nature of LLMs for simplicity in generating large-scale data.
>
> * In response to the suggestion, we conduct the experiments of path patching on the two-digit sentence templates: "{A1}{A2} + {B1}{B2} = ", and measure the causal effects on the averaged logit of both {C1} and {C2}. We find that the distribution of key heads remains analogous to the results on "{A} + {B} = ". This phenomenon reveals that the key heads responsible for attending to the numbers are also involved in two-digit addition. We hypothesize this is because one-digit addition serves as the fundamental computation *"unit"* for multi-digit addition.
>
> * To investigate the potential of extending the observed effects of one-digit addition to more addition formats, we conduct the knockout experiments on multi-digit integers (a1-a3), rational numbers (b1), and multiple addition operations (c1). The table shows one representative sample for each format and the accuracy change after knocking out the key heads in LLaMA2-7B.
>
> | settings | formats | samples | accuracy |
> |-------| ------- | ------- | ------- |
> |     a1    | two-digit  |   "35 + 42 = " |    -69%     |
> |     a2   | three-digit  |   "154 + 243 = " |     -72%    |
> |     a3  | four-digit  |   "1524 + 3463 = " |    -76%     |
> |     b1   | rational  |   "2.4 + 7.2 = " |     -70%    |
> |     c1  | multi-add  |   "42 + 21 + 15 = " |    -78%     |
>
> * Upon observation, it is evident that in settings a1-a3, the performance of samples containing multi-digit numbers significantly declines after knocking out the identified key heads based on one-digit addition. The decline becomes more pronounced when scaling up to larger numbers. This suggests that the perturbation effects on one-digit addition may accumulate and have a greater impact when applied to larger numbers.
> Furthermore, in settings b1 and c1, the decline in performance provides additional support for the hypothesis that the one-digit addition *"unit"* could lay the foundation for more complex addition operations.

---

> > ### Author Response · Authors · 2023-11-16
> > **Response to Reviewer 7p7z [part 2/2]**
> >
> > **Q2**: More experiments on other mathematical tasks:
> > > "The robustness of this study could be significantly enhanced if the authors were to conduct analogous experiments on subtraction, multiplication, and division."
> >
> > **Ans for Q2:**
> > Thanks a lot for bring up this issue! We conduct the path patching experiments on four mathematical tasks using the following representative templates.
> >
> > | tasks | template1 | template2 | template3 |
> > | ------- | ------- | ------- |  ------- |
> > |    addition   | "{A} + {B} = " |    "The sum of {A} and {B} is "   | "Question: What is {A} plus {B}? Answer: "|
> > |    subtraction   | "{A} - {B} = "|    "The difference between {A} and {B} is "   | "Question: What is {A} minus {B}? Answer: " |
> > |    multiplication   | "{A} * {B} = " |    "The product of {A} and {B} is "   | "Question: What is {A} times {B}? Answer: "|
> > |    division   | "{A} / {B} = "|    "The ratio between {A} and {B} is "   | "Question: What is {A} over {B}? Answer: " |
> >
> > * We find that: (i) the sparsity of key heads remains consistent across all four tasks (less than 1.0% of all heads). (ii) The key heads mainly distribute in the middle layers. The phenomena are analogous to the primary findings on the addition task (Section 4.1), demonstrating the potential of extending the observed effects of the addition task to other mathematical tasks.
> > * We compare the location of key heads across four mathematical tasks. An interesting finding is that the key heads used in "subtraction" and "addition" tasks overlapped significantly, as did the key heads used in "multiplication" and "division" tasks. Moreover, the four tasks share the heads (e.g., 13.11 and 12.22) that deliver the most significant effects, while they have task-specific heads that only emerge in its own task. These findings suggest that LLMs exhibit behavior aligned with human thinking to some extent, since "subtraction-addition" and "multiplication-division" are opposite mathematical operations.
> >
> > Thanks for your valuable suggestion, the above results and discussions have been added to the revision (Appendix B).
> >
> > **Q3**: More insights:
> > > "It would be great if authors have any additional insights regarding the points in weakness section."
> >
> > **Ans for Q3:**
> > Thanks for the inspiring comments in weakness section. We summarize two additional insights as follows:
> > * In the responses to **Q1**, we scale up the one-digit addition to more complex scenarios (e.g., multi-digit numbers, rational numbers, and multiple addition operations), and discover the accumulation effect based on one-digit addition. This may provide an explanation for why *"the ability of LLMs to do arithmetic quickly decreases with the increase of the number of digits"* [1, 2].
> > * In the responses to **Q2**, we extend the addition task to more mathematical tasks (e.g., subtraction, multiplication, division), and observe the symmetry of "addition-subtraction" and "multiplication-division". At a certain level, this may reveal the LLMs sharing the same spirit with humans.
> >
> > [1] Limitations of Language Models in Arithmetic and Symbolic Induction. In ACL 2023.
> > [2] Overcoming the Limitations of Large Language Models. In Towards Data Science 2023.

---

> ### Author Response · Authors · 2023-11-20
> **Welcome for more discussions (#7p7z)**
>
> Thanks for your valuable time in reviewing and constructive comments, according to which we have tried our best to answer the questions and carefully revise the paper. Here is a **summary of our response** for your convenience:
>
> - (1) **Addition task issues**: Following your constructive comments, we have conducted experiments on more number samples, including "multiple integers", "rational numbers", and "multiple addition operations". The results reveal that the potential of extending the scope of this work to multi-digit and non-trivial scenarios.
> - (2) **More math tasks**: Following your valuable suggestions, we have conducted experiments on more tasks of subtraction, multiplication and division. We also summarize the insightful phenomena of the shared key heads across the opposite tasks of "addition-subtraction" and "multiplication-division".
>
> We humbly hope our repsonse has addressed your concerns. If you have any additional concerns or comments that we may have missed in our responses, we would be most grateful for any further feedback from you to help us further enhance our work.
>
>
> Best regards
>
> Authors of #3226

---

> ### Author Response · Authors · 2023-11-21
> **Window for responsing and draft updating is closing**
>
> Dear Reviewer #7p7z,
>
> Thanks very much for your time and valuable comments. We understand you're busy. But as the window for responsing and paper revision is closing, would you mind checking our response ([a brief summary](https://openreview.net/forum?id=VpCqrMMGVm&noteId=ppKUir1vrs), and [details](https://openreview.net/forum?id=VpCqrMMGVm&noteId=Bt8RQxcX6u)) and confirm whether you have any further questions? We are very glad to provide answers and revision to your further questions.
>
> Best regards and thanks,
>
> Authors of #3226

---

> ### Comment · Reviewer_7p7z · 2023-11-21
> **Re: authors response**
>
> I thank the authors for their time and clarifying the points I raised. The supplementary experiments and insights provided by the authors can be of a significant value of the paper. I am revising my score from 5 to 6.

---

> > ### Author Response · Authors · 2023-11-22
> > **Thanks for increasing scores**
> >
> > Dear reviewer #7p7z,
> >
> > Thanks a lot for your prompt response despite such a busy period. We sincerely appreciate that you raise the score. Your valuable comments greatly help us in enhancing our work. If you have any further questions or comments, we are very glad to discuss more with you.
> >
> > Best regards
> >
> > Authors of #3226

---

### Official Review · Reviewer_K8c8 · 2023-11-01

**Soundness:** 3 good
**Presentation:** 3 good
**Contribution:** 2 fair
**Rating:** 6
**Confidence:** 4

**Summary:**

This paper investigates how three different language models (LLMs) perform on simple one-digit addition problems. The researchers generated 10,000 sample addition questions across 20 different formats (such as "42 plus 34 equals ___") to analyze. Through this analysis, they identified the most important attention heads involved in the addition calculations for each model. To confirm the importance of these heads, the researchers ablated them and used counterfactual examples, which showed a clear impact on loss when these heads were removed. Interestingly, only a very small number of attention heads were consistently involved in the addition across all the different question formats. Further examination showed these heads specifically focus on the numerical tokens in the input strings. The researchers replicated some of these findings with one-digit subtraction as well. The paper clearly maps out how a few key attention heads enable simple addition across different state-of-the-art LLMs.

**Strengths:**

Strengths:

- The language of the paper is concise and clear.
- The breadth and depth of the paper is excellent - specifically the use of 3 LLMs
(LLaMA2-7B, Qwen-7B and chatGLM2-6B), 20 question formats and 10K sample
questions.
- The rigorous nature of the paper is excellent - the claims re addition are confirmed via
detailed experimentation.
- The most significant finding is that a small number of attention heads are consistently
used by each model to perform one-digit addition across the various question formats.

**Weaknesses:**

Weaknesses:

- The paper (seems to) limit itself to one-digit addition and subtraction - reducing its scope
to a subset of addition and subtraction. The abstract should explicitly say that the scope
is one-digit integer addition.
-  The paper (seems to) limit itself to simple one-digit addition and subtraction (without
“carry over one” or “borrow one” examples - reducing its scope to a subset of addition
and subtraction. The abstract should explicitly say that the scope is simple one-digit
integer addition.
-  The paper does not explain how the attention heads (&/or MLP layer) actually perform
the addition calculation. This explanation is left for future work.
-  The paper touches on subtraction, showing similarities, but a detailed analysis is left for
future work.
-  A discussion of the differences in how each of the LLMs implement one-digit addition
would have been interesting e.g. do all the models use roughly the same number of attention heads to implement addition? If no differences were found, then this would be
an interesting finding in itself.
-  The small scope of this paper limits the reusability of this work.

**Questions:**

Questions:

- The addition examples seem to be “simple” one-digit integer addition with a one
character answer. There appear to be no “carry over one” examples in the test questions
e.g “5 plus 7 is equal to 1_”. If this is so, it reduces the findings scope to some
subclasses of addition.

- The subtraction examples all seem to be “simple” one-digit integer subtraction with a one
character answer. There appear to be no “borrow one” examples in the test questions
e.g “112 minus 5 is equal to 10_”. If this is so, it reduces the findings scope to some
subclasses of subtraction.

- The calculation of the subtraction question “{A} - {B} =” likely has two distinct calculation
algorithms: one for when A > B and one for when A < B. Do the authors think that this
explains the 52% performance drop when the addition attention heads are ablated?

---

> ### Author Response · Authors · 2023-11-16
> **Response to Reviewer K8c8 [part 1/2]**
>
> We would like to thank the reviewer for taking the time to review our work. We are grateful that you find our research to possess excellent breadth, depth, and rigorous nature, while also being presented in a concise and clear manner. Thank you for acknowledging our efforts to the significant findings. According to your valuable comments, we provide detailed feedback.
>
> **Q1**: The scope of this work:
> > "i) one-digit addition and subtraction - reducing its scope to a subset of addition and subtraction. The abstract should explicitly say that the scope is one-digit integer addition."
> > "ii) simple one-digit addition and subtraction (without “carry over one” or “borrow one” examples - reducing its scope to a subset of addition and subtraction. The abstract should explicitly say that the scope is simple one-digit integer addition."
> > "iii) The small scope of this paper limits the reusability of this work."
>
> **Ans for Q1:**
> **i)** Thanks for your constructive suggestion. Accordingly, we have conducted more experiments and added the results/explanations to our revision.
>
> * We have revised the scope in the Abstract to "simple one-digit integer addition" in the revision. We would like to note that the reason why we experiment on one-digit addition in the primary scope is that the adopted three LLMs (e.g., LLaMA2-7B) tokenize each digit individually (e.g., '42' is tokenized to '4' and '2'). We follow the one-digit nature of LLMs for simplicity in generating large-scale data.
>
> * In response to your kind suggestion, we conduct the experiments of path patching on the two-digit sentence templates: "{A1}{A2} + {B1}{B2} = ", and measure the causal effects on the averaged logit of both {C1} and {C2}. We find that the distribution of key heads remains analogous to the results on "{A} + {B} = ", albeit with different magnitude of the effect. This phenomenon reveals that the key heads responsible for attending to the numbers are also involved in two-digit addition. We assume this is because one-digit addition serves as the fundamental computation *"unit"* for multi-digit addition, demonstrating the potential of extending the observed effects of one-digit addition to multi-digit addition.
>
> **ii)** Thank you for the insightful comment. Following the knockout experiments in Section 4.3, we first generate 200 samples that need to carry over one or borrow one (e.g., "17 + 9 = " and "17 - 9 = "). Then we knock out the identified key heads in addition, resulting in a wrong prediction on over 150 samples. We assume this is because the key heads attend to the addends "7" and "9", regardless of whether it needs to carry over one or borrow one.
>
> **iii)** We agree that a larger scope is important for the reusability. The above studies suggest the potential of extending the scope of this work to multi-digit and non-trivial scenarios. More analyses and discussions will be updated in the revision.
>
> We believe these novel observations would benefit a lot to the quality of our paper. Thanks again for your constructive suggstions.
>
> **Q2**: More explanation:
> > "explain how the attention heads (&/or MLP layer) actually perform the addition calculation"
>
> **Ans for Q2:**
> Thank you for bringing up the insightful comment!
> * We agree that delving into the *"causal mechanism"* of addition calculation is a quite promising direction. In this work, our primary contribution lies in analyzing the *"causal effect"* of component mediators to the model behavior. We also noticed that recent research [1, 2] tried to explain MLPs responsible for performing computations and retrieving facts. However, it necessitates thorough and rigorous analysis to make the detailed process in MLPs human-understandable, and interpret the collaboration between attention heads and MLPs. We believe that making the calculation fully understandable to humans is an intriguing direction that warrants sustained research efforts.
> * In response to the comment, we added one experiment of path patching to measure the causal effect of each MLP layer. The results indicate that: (i) the early MLP layers 0-13 before the key heads (e.g., 13.11) have a slight effect on the output (approximately $\pm$0.5%). (ii) The late MLP layers 14-31 after the key heads exhibit a much larger effect (approximately $\pm$10.0%). These findings may reveal that the key heads are responsible for attending to number tokens, while the following MLP layers process the number tokens.
>
>
> [1] Locating and Editing Factual Associations in GPT. In NeurIPS 2022.
> [2] Transformer Feed-Forward Layers Are Key-Value Memories. In EMNLP 2021.

---

> > ### Author Response · Authors · 2023-11-16
> > **Response to Reviewer K8c8 [part 2/2]**
> >
> > **Q3**: Detailed analysis on subtraction:
> > > "The paper touches on subtraction, showing similarities, but a detailed analysis is left for future work."
> >
> > **Ans for Q3:**
> > Thanks for your constructive suggestion.
> > * We conduct the experiment of path patching based on the subtraction sentences. We find that the identified key heads in the subtraction task are almost the same to those in the addition task, albeit with different magnitude of the effect. This phenomenon could reveal the similar key head *"location"* in addition and subtraction.
> > * Moreover, as we show the attention pattern of key heads, we observe that these heads particularly attend to the number tokens regardless of whether they are given addition or subtraction sentences. This phenomenon could reveal the similar key head *"behavior"* in addition and subtraction.
> >
> > The above studies further explain why knocking out the key heads identified in addition could affect the accuracy in performing subtraction. We have add the above discussions in the revision (Appendix B).
> >
> > **Q4**: Discussion of different LLMs:
> > > "A discussion of the differences in how each of the LLMs implement one-digit addition would have been interesting"
> >
> > **Ans for Q4:**
> > Thanks for your inspiring comment:
> > * We find that the quantity of key heads in different LLMs is different from each other. This could be attributed to their different architecture, model size, learning paradigm, training corpus, and more.
> > * We also noticed that the adopted three LLMs show different ability in understanding the input context. Given the input addition sentences (e.g., "The sum of {A} and {B} is "), we gather the averaged prediction probabilities of number tokens 1-9. The results are: LLaMA2-7B=99.3%, Qwen-7B=97.6%, and chatGLM2-6B=94.7%. The higher probability of LLaMA2-7B shows that it generates number tokens with a greater confidence level, demonstrating a better understanding of the input context. We assume this explains why LLaMA2-7B has more key heads in performing the addition task.
> >
> > **Q5**: Question for subtraction implementation:
> > > "The calculation of the subtraction question “{A} - {B} =” likely has two distinct calculation algorithms"
> >
> > **Ans for Q5:**
> > Thanks for your careful reviews. We agree that the calculation algorithms of "{A} > {B}" and "{A} < {B}" are distinct. As mentioned in our response to Q3, we hypothesize that the decrease in performance could be attributed to the shared key heads in both subtraction and addition tasks. The identified key heads in the addition task are also involved in the inner workings of subtraction. More discussions have been incorporated in our revision (Appendix B).
> >
> > **Q6**: Question for examples:
> > > "i) The addition examples seem to be “simple” one-digit integer addition with a one character answer."
> > > "ii) The subtraction examples all seem to be “simple” one-digit integer subtraction with a one character answer."
> >
> > **Ans for Q6:**
> > Thanks for bring up the concern. We conduct the experiments in the responses to **Q1**, and illustrate the scope of this work could extend to multi-digit and non-trivial scenarios.

---

> > > ### Comment · Reviewer_K8c8 · 2023-11-21
> > > **Response to authors**
> > >
> > > Thank you for taking the time to respond to my questions and comments. I find the responses helpful and i think the papers quality can improve with further experiments and results. I will maintain my score.

---

> > > > ### Author Response · Authors · 2023-11-21
> > > > **Thanks for your feedback.**
> > > >
> > > > Dear Reviewer K8c8,
> > > >
> > > > Thank you very much for the feedback. We are pleased to hear that our responses are helpful. We will incorporate the further experiments and results into our revision.
> > > >
> > > > Best regards
> > > >
> > > > Authors of #3226

---

> ### Author Response · Authors · 2023-11-20
> **Welcome for more discussions (#K8c8)**
>
> Thanks for your valuable time in reviewing and constructive comments, according to which we have tried our best to answer the questions and carefully revise the paper. Here is a **summary of our response** for your convenience:
>
> - (1) **Work scope issues**: Following your constructive comments, we have conducted experiments on more examples, including "multi-digit", "carry over one", and "borrow one". The results reveal that the potential of extending the scope of this work to multi-digit and non-trivial scenarios.
> - (2) **More discussions**: Following your valuable suggestions, we have provided more discussions about the interpretation of attention heads and MLPs, the analysis on the symmetry between addition and subtraction, the comparison of key heads between different LLMs, and the reason why the performance in subtraction decreases.
>
> We humbly hope our repsonse has addressed your concerns. If you have any additional concerns or comments that we may have missed in our responses, we would be most grateful for any further feedback from you to help us further enhance our work.
>
>
> Best regards
>
> Authors of #3226

---

> ### Author Response · Authors · 2023-11-21
> **Window for responsing and draft updating is closing**
>
> Dear Reviewer #K8c8,
>
> Thanks very much for your time and valuable comments. We understand you're busy. But as the window for responsing and paper revision is closing, would you mind checking our response ([a brief summary](https://openreview.net/forum?id=VpCqrMMGVm&noteId=1ecAeVExTs), and [details](https://openreview.net/forum?id=VpCqrMMGVm&noteId=LyKFOSwBNg)) and confirm whether you have any further questions? We are very glad to provide answers and revision to your further questions.
>
> Best regards and thanks,
>
> Authors of #3226

---

### Official Review · Reviewer_jEqH · 2023-11-01

**Soundness:** 3 good
**Presentation:** 3 good
**Contribution:** 3 good
**Rating:** 6
**Confidence:** 3

**Summary:**

Three workings of models, LLaMA2-7B, Qwen-7B, and  chatGLM2-6B, are interpreted using the path patching method (initially introduced in [1], which is an interoperability method rooted in causal intervention) on tasks involving mathematical addition and subtraction. The authors create various datasets for this purpose. They find that only a small number of attention heads are responsible for reasoning.

This represents a good effort to interpret large language models using path patching and mean ablation and it is the first paper where mathematical addition is interpreted in this way.

[1] https://openreview.net/pdf?id=NpsVSN6o4ul

**Strengths:**

- a timely topic is treated, how models that are used in practice perform mathematical addition and subtraction
- a large number of figures that show how attention heads are activated on concrete examples help to make the paper readable

**Weaknesses:**

- The authors didn't include, as related work, some publications that also deal with mathematical reasoning, such as [1]
- studying only mathematical addition and subtraction seems restrictive. I do note that the authors state at the end however: "_A more thorough study on the subtraction task as well as the validation on more computation tasks (e.g., multiplication and division, etc.) is left for future work._"

[1] https://arxiv.org/pdf/2305.08809.pdf

**Questions:**

-Since addition and subtraction are opposite mathematical operations, is there some kind of similar symmetry observable on the level of attention heads?

**Details Of Ethics Concerns:**

(not applicable)

---

> ### Author Response · Authors · 2023-11-16
> **Response to Reviewer jEqH**
>
> We would like to thank the reviewer for taking the time to review our work. We appreciate that you find our work represents a good effort in **a timely topic** and is easy to follow with concrete examples. Thank you for acknowledging our research as the first paper with significant findings of interpreting mathematical addition in LLMs. According to your valuable comments, we provide detailed feedback.
>
> **Q1**: Related work:
> > "some publications that also deal with mathematical reasoning, such as [1]."
>
> **Ans for Q1:**
> Thanks for bringing attention to this outstanding work [1], we have added a discussion in the Background section. The studies in [1] scale the methods from causal abstraction to understand how Alpaca (7B) follows a particular instruction: "Please say yes only if it costs between [X] and [Y] dollars, otherwise no.", where it needs to compare the input value with the lower bound [X] and the upper bound [Y]. Different from the task of *number comparison*, we focus on the *number addition* behavior of LLMs and attempt to make their inner workings more understandable to humans.
>
> [1] Interpretability at Scale: Identifying Causal Mechanisms in Alpaca. In ArXiv preprint 2023.
>
> **Q2**: Extend to more tasks:
> > "studying only mathematical addition and subtraction seems restrictive."
>
> **Ans for Q2:**
> Thanks for your constructive suggestion. We conduct experiments on four mathematical tasks using the following representative templates.
>
> | tasks | template1 | template2 | template3 |
> | ------- | ------- | ------- |  ------- |
> |    addition   | "{A} + {B} = " |    "The sum of {A} and {B} is "   | "Question: What is {A} plus {B}? Answer: "|
> |    subtraction   | "{A} - {B} = "|    "The difference between {A} and {B} is "   | "Question: What is {A} minus {B}? Answer: " |
> |    multiplication   | "{A} * {B} = " |    "The product of {A} and {B} is "   | "Question: What is {A} times {B}? Answer: "|
> |    division   | "{A} / {B} = "|    "The ratio between {A} and {B} is "   | "Question: What is {A} over {B}? Answer: " |
>
> - Our experimental results show that:
>   - The sparsity of key heads remains consistent across all four tasks (less than 1.0% of all heads);
>   - The key heads mainly distribute in the middle layers.
> The phenomena are analogous to the primary findings on the addition task (Section 4.1), demonstrating the potential of extending the observed effects of the addition task to other mathematical tasks.
> * We compare the location of key heads across four mathematical tasks. An interesting finding is that the key heads used in "subtraction" and "addition" tasks overlapped significantly, as did the key heads used in "multiplication" and "division" tasks. Moreover, the four tasks share the heads (e.g., 13.11 and 12.22) that deliver the most significant effects, while they have task-specific heads that only emerge in its own task. These findings suggest that LLMs exhibit behavior aligned with human thinking to some extent, since "subtraction-addition" and "multiplication-division" are opposite mathematical operations.
>
> The above results and discussions have been incorporated into the revision (Apendix B).
>
> **Q3**: Analysis of addition and subtraction tasks:
> > "is there some kind of similar symmetry observable on the level of attention heads?"
>
> **Ans for Q3:**
> Thanks a lot for bringing up this insightful question!
> * In the above experiments for **Q2**, we find that the identified key heads in the addition task are almost the same to those in the subtraction task, albeit with different magnitude of the effect. This phenomenon may reveal the symmetry of key head *"location"* in addition and subtraction.
> * Moreover, as we show the attention pattern of key heads, we observe that these heads particularly attend to the number tokens regardless of whether they are given addition or subtraction sentences. This phenomenon may reveal the symmetry of key head *"behavior"* in addition and subtraction.

---

> ### Author Response · Authors · 2023-11-20
> **Welcome for more discussions (#jEqH)**
>
> Thanks for your valuable time in reviewing and constructive comments, according to which we have tried our best to answer the questions and carefully revise the paper. Here is a **summary of our response** for your convenience:
>
> - (1) **Related work issues**: Following your constructive comments, we have discussed related works in mathematical reasoning (Alpaca) to highlight our novelty. And we also add these discussions into our revision to enhance our work.
> - (2) **Experiment task issues**: Following your valuable suggestions, we have conducted experiments on more tasks of subtraction, multiplication and division. We also summarize two aspects of the similar symmetry in addition and subtraction.
>
> We humbly hope our repsonse has addressed your concerns. If you have any additional concerns or comments that we may have missed in our responses, we would be most grateful for any further feedback from you to help us further enhance our work.
>
>
> Best regards
>
> Authors of #3226

---

> ### Author Response · Authors · 2023-11-21
> **Window for responsing and draft updating is closing**
>
> Dear Reviewer #jEqH,
>
> Thanks very much for your time and valuable comments. We understand you're busy. But as the window for responsing and paper revision is closing, would you mind checking our response ([a brief summary](https://openreview.net/forum?id=VpCqrMMGVm&noteId=cxle2Siqb5), and [details](https://openreview.net/forum?id=VpCqrMMGVm&noteId=PhYFDwLBxp)) and confirm whether you have any further questions? We are very glad to provide answers and revision to your further questions.
>
> Best regards and thanks,
>
> Authors of #3226

---

> > ### Comment · Reviewer_jEqH · 2023-11-23
> > **Reviewer answer**
> >
> > I thank the authors for their efforts in revising the paper.
> >
> > It reads much better now. (But please be careful, there are still some typos in some sections, e.g. "GPT2-samll" in the background sections.) I will maintain my score.

---

> > > ### Author Response · Authors · 2023-11-23
> > > **Thanks for your feedback.**
> > >
> > > Dear Reviewer jEqH,
> > >
> > > Thank you very much for the feedback. Your valuable suggestions improve the quality of this paper. We will carefully correct the typos in our revision.
> > >
> > > Best regards
> > >
> > > Authors of #3226

---

### Meta-Review · Area_Chair_jwGk · 2023-12-05

**Metareview:**

This work investigates the operation of several LLMs on the task of one-digit addition.
The authors apply tools from the mechanistic interpretability literature to investigate the attention layers.
Their main findings are:
* There exist a small set of attention heads (0.5%) which are crucial to performance, meaning “knocking-out” these heads destroys addition performance.
* These key heads tend to attend to the relevant numeric tokens in the input (across various input formats).

Reviewers agreed that the topic is timely, and that the authors have made good efforts towards their goal. The paper is overall well-written, and the experiments are clear. The authors added several experiments and re-wrote certain parts in the rebuttal stage, which improved the quality of the paper.

However, reviews were not enthusiastic about the strength and significance of the results. Reviewers observed the following main weaknesses. After reading the paper and the rebuttals, I agree with these myself.
* The scope is very limited (1-digit addition, with a few types of prompts, pretrained models). It’s unclear what to take away from this setting, and there is not much discussion of the motivation to study this particular setting.
* The conclusions are fairly weak: the paper does not propose nor investigate the mechanisms used for addition. They observe that several attention heads are crucial, and that these heads attend to the relevant parts of the input. However, they do not deeply investigate what these heads actually do, and how they interact with the MLP layers. The authors acknowledged these limitations in their rebuttals, but the paper still contains over-claims of significance (e.g. “The above results reveal a distinct working mechanism of the discovered key heads on number tokens, and verify the aforementioned hypotheses”).

Unfortunately, while the results are interesting, they are not strong enough for acceptance to ICLR for the above reasons. Thus I recommend rejection.

I appreciate that understanding / interpretability of LLMs is scientifically difficult, and it can be useful to start with a small scope, and use heuristic techniques like knockouts and path patching to gain insight. Ultimately however, there was not enough insight gleaned from these experiments for this work to appear in ICLR.

**Justification For Why Not Higher Score:**

The strength and significance of the results are lacking.

**Justification For Why Not Lower Score:**

N/A

---

### Decision · Program_Chairs · 2024-01-16

Reject